# Structural basis for lamin assembly at the molecular level

Jinsook Ahn [1,4], Inseong Jo [1,4], So-mi Kang[2], Seokho Hong [1], Suhyeon Kim[1], Soyeon Jeong [1], Yong-Hak Kim [3], Bum-Joon Park[2] & Nam-Chul Ha [1]

Nuclear structure and function are governed by lamins, which are intermediate filaments that mostly consist of α-helices. Different lamin assembly models have been proposed based on low resolution and fragmented structures. However, their assembly mechanisms are still poorly understood at the molecular level. Here, we present the crystal structure of a long human lamin fragment at 3.2 Å resolution that allows the visualization of the features of the full-length protein. The structure shows an anti-parallel arrangement of the two coiled-coil dimers, which is important for the assembly process. We further discover an interaction between the lamin dimers by using chemical cross-linking and mass spectrometry analysis. Based on these two interactions, we propose a molecular mechanism for lamin assembly that is in agreement with a recent model representing the native state and could explain pathological mutations. Our findings also provide the molecular basis for assembly mechanisms of other intermediate filaments.

[1] Department of Agricultural Biotechnology, Center for Food Safety and Toxicology, Center for Food and Bioconvergence, and Research Institute for Agriculture and Life Sciences, CALS, Seoul National University, Seoul 08826, Korea. [2] Department of Molecular Biology, College of Natural Science, Pusan National University, Busan 46241, Korea. [3] Department of Microbiology, Catholic University of Daegu School of Medicine, Daegu 38430, Korea. [4] These authors contributed equally: Jinsook Ahn, Inseong Jo Correspondence and requests for materials should be addressed to N.-C.H. (email: hanc210@snu.ac.kr)

Intermediate filament (IF) proteins form flexible fibrous structures that provide vital mechanical support in higher eukaryotic cells[1,2]. All IF proteins have three domains: an unstructured N-terminal head, a central α-helical rod, and non-helical C-terminal tail domains[2–4]. The central rod domain is further divided into three coiled-coil segments (coil 1a, 1b, and 2) and two flanking linkers (L1 and L12). The central rod domains have a high propensity to form coiled-coil dimers, which are further assembled into a 10-nm-thick filament in cells in several hierarchical steps in most IF proteins[5].

The nuclear envelope structures are the hallmark of all eukaryotic cells, and the intact structures are important for the proper functioning of the nucleus and the cells. Nuclear lamins are IF proteins that play an essential role in maintaining the nuclear envelope structure[6,7]. They have been implicated in diverse cellular processes, including mitosis, chromatin organization, DNA replication, and transcription[8–10]. Many lamin mutations have been found to be closely related to various human diseases, including muscular dystrophy and Hutchinson Gilford progeria syndrome[11–13].

Compared with the canonical IF proteins, such as vimentin and keratin, the central rod domain of lamins has a longer coil 1b, and the non-helical C-terminal tail region contains an additional immunoglobulin (Ig)-like domain[2,14]. The fundamental soluble unit of lamin is a dimer, which is different from the tetrameric vimentin[14,15]. The lamin assembly is formed by longitudinal and lateral association based on the coiled-coil dimers[16,17]. Initially, many researchers studied lamin using the refolded proteins, proposing an assembly model with 2–4 nm overlap between the dimers by emphasizing the head-to-tail interaction between the dimeric units of lamin[14]. Several fragments structures, together with vimentin fragment structures, have been determined, and full-length models of vimentin in the tetrameric arrangement have been proposed by combining the fragment structures and binding mapping with the electron paramagnetic resonance spectroscopy[17,18]. Recently, a cryo-electron tomography (cryo-ET) study revealed that lamins form 3.5-nm-thick filaments[16] that are remarkably different from other canonical 10-nm thick IF proteins and the proposed assembly model of lamin[3,16,19,20]. The Ig-like domains of lamins were decorated at both sides of the filament with an interval of ~20 nm and various cross-sections[16].

The lamin filaments make up the three-dimensional meshwork underlying the inner nuclear envelope[10,21–23]. The intrinsic structural flexibility and self-aggregative properties of lamin have allowed determination of only low-resolution electron microscopy (EM) structures and crystal structures of short fragments[24–26]. To date, their assembly mechanisms at the molecular level are poorly understood[16,26–28]. In this study, we determined the crystal structure of an N-terminal half fragment of the human lamin A/C, which was stably expressed and amenable to studies for the seamless full-length structures of lamin. This study provides a structural basis for how the lamin filament is assembled with two interactions, giving insights into the assembly mechanisms of other IFs.

## Results

**Structure determination of the lamin fragment**. We determined the crystal structure of an N-terminal half fragment of human lamin A/C (residues 1–300; referred to as 'lamin 300 fragment' in this study) at a resolution of 3.2 Å. This spans the N-terminal head, coil 1a, L1, coil 1b, L12, and the first half of coil 2 (Fig. 1a and Supplementary Table 1). The asymmetric unit consists of four protomers (chains A–D), which are bundled by an anti-parallel pair of parallel helical dimers (chain A:B and C:D). The four helical bundle structure showed the α-helical conformations in an ~38-nm-long structure (Fig. 1b).

The analyses of the two helical dimers using the programme TWISTER (Supplementary Fig. 1a)[29] showed two different super-helical structures with regular α-helical regions. A typical left-handed coiled-coil conformation was found in most of the coil 1a and the N-terminal four-fifths of coil 1b (Supplementary Fig. 1a), in which hydrophobic residues were regularly found at the heptad positions a and d (Fig. 2a and Supplementary Fig. 1b; highlighted in yellow). Remarkably, significant bending was found in the regions near L1, probably caused by the crystal packing interactions (Fig. 2b left). The junction between coil 1a and L1 (residues 65–69) showed a short α-helical segment without an inter-helical hydrophobic interaction (Fig. 2). Linker L1 and an adjacent coil 1b segment (residues 70–84) showed the hendecad pattern[30,31], which induces the formation of an untwisted helical section (Fig. 2 and Supplementary Fig. 1). The hendecad pattern is composed of 11 amino-acid repeats, where the residues at the a, d, and h positions are involved in the inter-helical hydrophobic interactions (Fig. 2a and Supplementary Fig. 1b; highlighted in magenta).

In the helical dimers, coil 1b consists of the 18 N-terminal heptad repeats (residues 79–204) forming a left-handed coiled-coil structure (~22 nm; one and a half super-helical turn) and the two C-terminal hendecad repeats (residues 204–222) forming an untwisted helical section (Fig. 2 and Supplementary Fig. 1). The hendecad repeat region in coil 1b is continuous to the N-terminal half of L12 (residues 223–230; Fig. 2a and Supplementary Fig. 1b), unlike previous predictions proposing a flexible hinge conformation[17,26]. The untwisted helical section is bent asymmetrically near the junction between L12 and coil 2 (Fig. 2 and Supplementary Fig. 1b), leading to mismatched inter-helical hydrophobic interactions (Supplementary Fig. 2a). Interestingly, the C-terminal end of the lamin 300 fragment interacts with the symmetry-related molecules through hydrophobic interactions, as observed in the coil 2 structure of vimentin[32] (Supplementary Fig. 2).

**Anti-parallel interaction between the two coiled coils**. The tetrameric arrangement in the asymmetric unit is noteworthy. Coil 1b and its flanking linker segments (residues 67–221) form the anti-parallel contacts between coiled-coil dimers, and the remaining parts (coil 1a and coil 2) are bifurcated from both sides of the contacting region (Fig. 3a). The anti-parallel contact region is subdivided into a central compartment and two side compartments, and each compartment is composed of a pair of half super-helical turns of two α-helical dimers (Fig. 3a). The two side compartments with the internal pseudo-2-fold symmetry exhibit cross-sectional variation. At the central compartment, the four α-helices form two dimers arranged in juxtaposition (Fig. 3a–c). The cross-section in the middle region is ~3.5 nm in width, which is similar to the thickness of the lamin filament model determined by cryo-ET[16] (Fig. 1b).

Structural comparison of the anti-parallel contact region of lamin to the vimentin coil 1b structure depicting the A11 interaction revealed a striking structural similarity between the side compartments of lamin and the vimentin coil 1b (Fig. 3a). We further noticed that the insertion sequence of coil 1b of lamin on the sequence alignment to vimentin represents the central compartment of the anti-parallel contact region (Supplementary Fig. 3). Each side compartment has two hydrophobic patches for the anti-parallel inter-dimer interaction (Fig. 3a, b, d, e). The inter-dimer interactions were not mediated by residues at the a, d, and h positions in the hendecad repeat regions, and neither was in vimentin (Supplementary Figs. 3 and 4). In contrast, only slight contacts were found in the central compartment (Fig. 3b, c). At the junctions between the central and side compartments, each

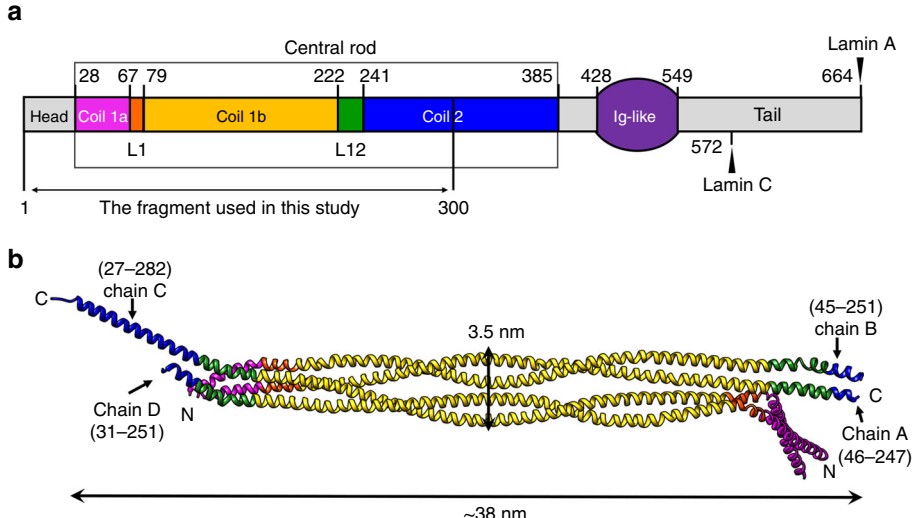

**Fig. 1** Structure of the asymmetric unit of the human lamin A/C fragment. **a** Structural organization of human lamin A/C, consisting of the head, central rod, Ig-like, and C-terminal tail domains. The central rod domain is divided into sub-domains (coil 1a, L1, coil 1b, L12, and coil 2). The start and end residues of coils 1a, 1b, and 2 are labelled with different C-termini in lamin A and lamin C. See also Supplementary Table 1. **b** The asymmetric unit in the crystal of an N-terminal fragment (residues 1–300) of lamin A/C consists of four chains: A–D. The ordered region in the crystal structure is indicated. Each subdomain is coloured according to the colour code in **a**

dimer is twisted by 90°, resulting in a parallel arrangement without direct contacts (Fig. 3a). From these observations, we concluded that the anti-parallel interaction between the two coiled-coil dimers in the asymmetric unit of lamin corresponds to the 'A11 interaction' of vimentin[18,33–35] (Fig. 3a and Supplementary Fig. 4). Thus, we referred to the anti-parallel contact between the two dimers of lamin as an 'A11 interaction' in this study.

**Importance of the lamin A11 interaction for the lamin assembly process.** It is known that lamin proteins exist in the dimeric form in solution before forming the filament[15,23]. To probe the A11 interaction of lamin during the filament formation, we noted Ala146 at the centre of the anti-parallel coiled-coil dimers, which would be close enough to make a disulfide bond if it is changed to cysteine (Fig. 3b, c). We first observed that the A146C variant protein of the lamin 300 fragment formed a disulfide bond in SDS-PAGE, suggesting that the tetrameric arrangement is formed in solution (Fig. 3c, d).

We next compared the molecular masses of the wild-type and A146C variant of the lamin 300 fragment using size exclusion chromatography-multiangle light scattering (SEC-MALS). The A146C variant protein partially formed a larger complex indicating a tetramer, while the wild-type fragment mostly remained as a dimer. Moreover, the larger tetrameric complex was shifted into the dimeric size in the presence of a reducing agent (Fig. 3d). These results indicate that Ala146 makes a close contact between two dimers during the assembly process, as predicted from the crystal structure, and further suggest that the tetrameric lamin formed by the A11 interaction represents a snapshot of the assembly processing.

**Enhanced binding of coil 2 by the A11 interaction.** We attempted to find the binding partners of the lamin 300 fragment with four candidate fragments, as shown in Fig. 4a. We performed a binding assay using the A146C variant of the lamin 300 fragment that makes more stable A11 interactions than the wild type lamin 300 fragment. Only the F3 fragment (residues 250–400) was bound to the A146C variant of the lamin 300 fragment (Fig. 4b). The F3 fragment is roughly

matched to coil 2 (residues 240–385) without the N-terminal ten residues. The binding of the coil 2 fragment was weakened when the disulfide bond was reduced in the A146C variant of the lamin 300 fragment, and the wild type of the lamin 300 fragment also bound to the coil 2 fragment with a weaker affinity than the A146C variant of the lamin 300 fragment (Fig. 4c). These results showed that coil 2 is bound to the lamin 300 fragment, and the binding affinity is increased by the A11 interaction.

**The eA22 interaction.** To identify the binding sites between the lamin 300 fragment and the bound coil 2 fragment, we conducted a chemical cross-linking assay, followed by MS/MS analysis. Since both fragments contain many lysine residues without any cysteine residue, we introduced a cysteine residue at the Arg388 site in the C-terminal end region of a shorter coil 2 fragment (residues 286–400). The cysteine residue would be covalently linked to a lysine residue of the lamin 300 fragment via sulfo-SMPB [sulfo-succinimidyl 4-(N-maleimidophenyl) butyrate] cross-linker if the coil 2 fragment is bound to the lamin 300 fragment. The treatment of the cross-linker at low concentration (2.5 μM) built up specific conjugate bands (~52 and ~30 kDa) on SDS-PAGE (Fig. 5a). The MS/MS analysis of the conjugate band (~52 kDa) indicated that the Cys388 site of coil 2 is near Lys171 in the lamin 300 fragment in the highest proportion among all modified lysine residues at each position (Fig. 5b and Supplementary Table 2). It is also possible that Lys180 and Lys181, which are close to the Lys171 site, are involved in this eA22 interaction if the length of cross-linker and surface geometry of the coiled-coil are considered. These results indicated that the entire coil 2 dimers make an extensive anti-parallel interaction with each other (Fig. 5c). We call this interaction an eA22 (extensive anti-parallel binding between coil 2s) interaction in this study and is distinguished from the 'A22 interaction' of vimentin, where only the C-terminal half parts of coil 2 (called coil 2b) are involved[36]. In addition, the ~30 kDa conjugate band was revealed to be the coil 2 fragment dimer.

Since the eA22 interaction was enhanced by the A11 interaction, the eA22 interaction should be considered in terms of the tetramer made by the A11 interaction. We structurally

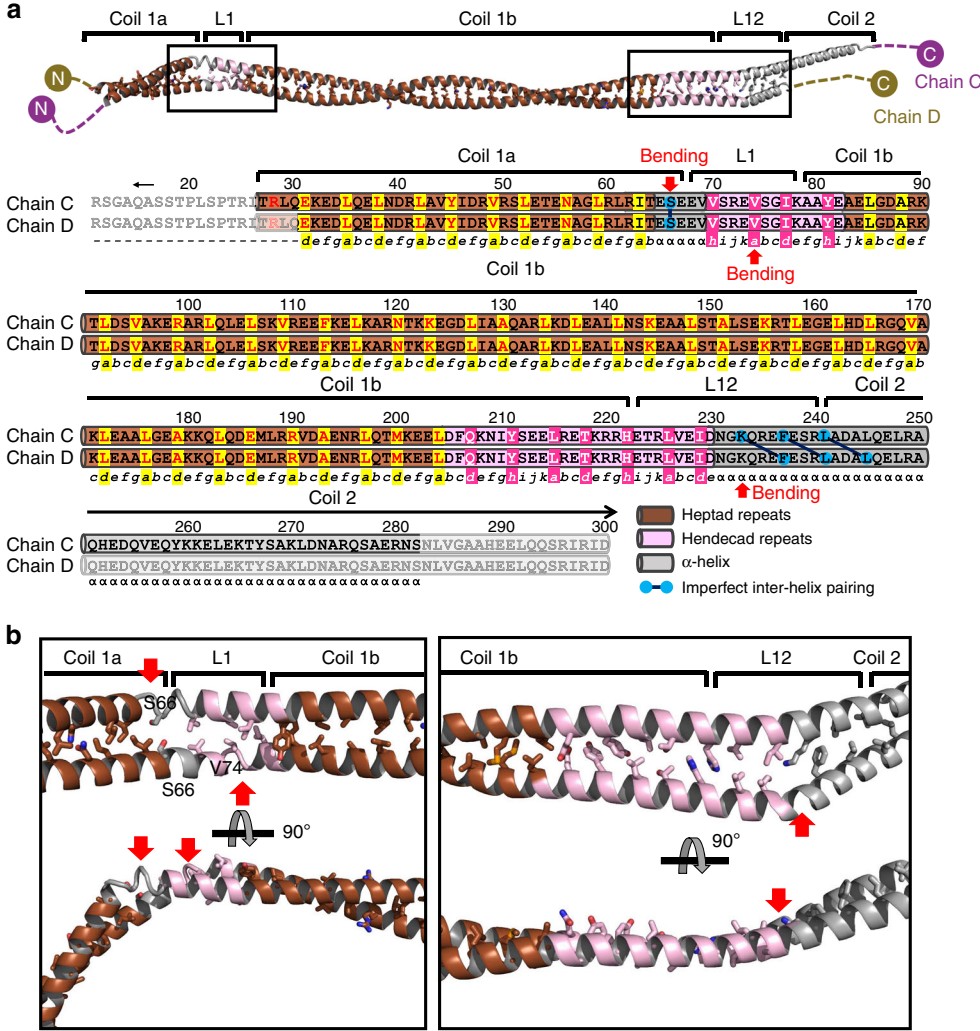

**Fig. 2** Structural-based superhelicity and sequence periodicity of the lamin C:D dimer. **a** Structure-based analysis of the sequence periodicity of the heptad (shaded in brown or yellow) or hendecad (shaded in pink or magenta) repeats. The phases of the heptad (abcdefg) and hendecad (abcdefghijk) repeats are described below the sequences. The hydrophobic residues (a, d, and h) involved in the inter-helical interaction are highlighted by shadowing. If no inter-helical interaction occurs, α is used instead. Bends in the α-helices are indicated by red arrows. The mismatched interaction between two α-helices is highlighted with linked blue circles. The disordered regions are masked with semi-transparent boxes. The analyses of the A:B dimer are provided in Supplementary Fig. 1b. **b** Inter-helical hydrophobic interactions of the linkers L1 and L12 proximal regions in the C:D dimer. Heptad repeat periodicity exhibiting the left-handed coiled-coil structure is coloured brown, whereas the hendecad repeat periodicity in the untwisted α-helical dimer is coloured pink. Grey α-helices indicate an unmatched hydrophobic interaction between two α-helices in the dimer. Hydrophobic residues associated with inter-helical interactions in the L1 and L12 proximal regions are shown in stick representations in the orthogonal views. The bending in the α-helices is indicated by red arrows

evaluated the possible overlapping region for the coil 2 region in the presence of the A11 and eA22 interactions. In the tetrameric arrangement by the A11 interaction, we noted ~6-nm-long spaces between the bifurcated coil 1a and coil 2 on both sides (Fig. 6). The bifurcated coils appear to hold a long α-helix or coiled-coil dimer like a crocodile clip if they are stretched from the core anti-parallel contact part (Fig. 6b), which would be allowed by the conformational flexibility at the linker segments (L1 and L12; Fig. 2 and Supplementary Fig. 1). We next noted ~4-nm-long grooves in the tetrameric arrangement at each side compartment in the anti-parallel contact part (Fig. 6a). The grooves would accommodate a longer structural motif together with the bifurcated coils, since they are continuous to the space between the bifurcated coil 1a and coil 2, making a 10-nm-long binding motif (Fig. 6b). The groove and space would provide the major part of the long overlapping region with the 6 α-helical intersection.

**Proposed lamin assembly model at the molecular level**. To build the assembly model of the lamin filament, we started from a 54-nm-long coiled-coil dimer model of the central rod domain by combining this structure with fragmented structures[25,26,32] (Fig. 7a). The missing parts (residues 283–385) were presumed by the typical coiled-coil structure with a rise per residue of 1.5 Å. The A11 interactions were applied between coiled-coil dimers, leading to the formation of a 72-nm-long tetramer with 19-nm-long overhangs of coil 2 on both sides (Supplementary Fig. 5a). Sequentially or simultaneously, coil 2 of the coiled-coil dimer was inserted to the tetramer by the eA22 interaction with ~14 nm-long overlapping, including 10-nm-long binding motif. This assembly resulted in ~40-nm-long intervals between the C-terminal ends of two coiled-coil dimers in the same direction (Fig. 7b and Supplementary Fig. 5b). Due to the twofold symmetry in the four lamin subunits, an interval averaging ~20 nm is generated between the Ig-like domains attached to coil 2 (Fig. 7c).

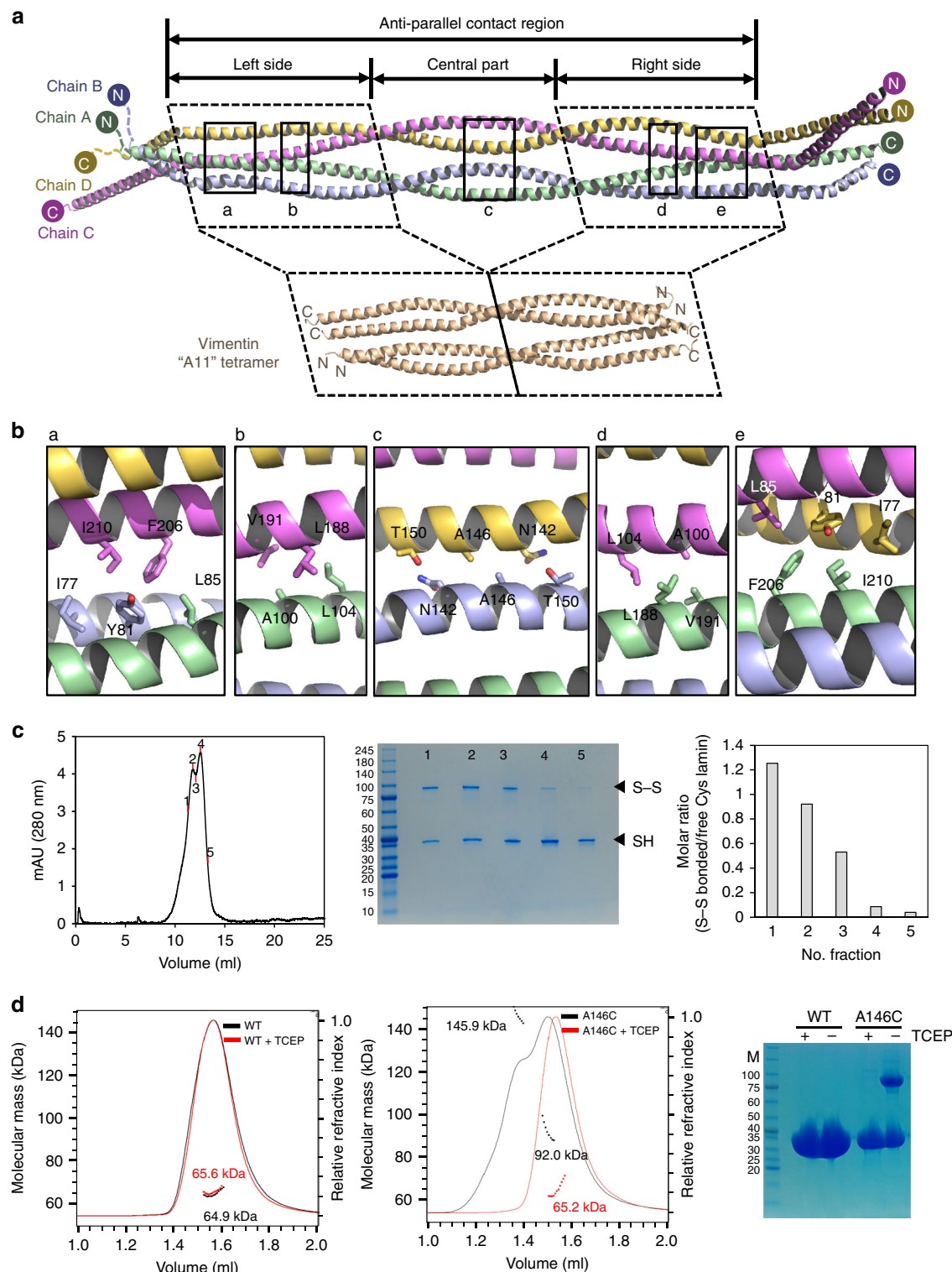

We believe that the A11 and eA22 interactions would occur simultaneously and synergistically for the further extensions of the lamin filament (Fig. 7b, c). This assembly model is well matched to the previous cryo-ET structure representing the native state, including the features of different cross-section shapes and the interval between the adjacent Ig-like domains[16].

**Laminopathies correlated with A11 and eA22 interactions.** We noted that many genetic mutations causing laminopathies were

mapped on the interfaces of the A11 and eA22 interactions, suggesting their importance in the formation of a functional nucleus (Supplementary Fig. 6a). To gain insights into the laminopathies at the molecular level, we selected two mutations Y45C and L59R, which are related to muscular dystrophy[37,38] and dilated cardiomyopathy[13,39,40] respectively. We compared the binding affinity of the lamin 1–300 fragments to the C-terminal part of coil 2 in low and high salt buffers containing 50 or 150 mM NaCl (Fig. 4c), where the eA22 interaction was strengthened in the low salt buffer more than that in a high salt buffer (Fig. 4c).

**Fig. 3** A11 interaction of the lamin 300 fragment. **a** Structural comparison of the tetrameric structures of the lamin 300 fragment (this study, residues 75–213) and vimentin coil 1b (PDB code: 3UF1[18], residues 144–251). Top panel, the anti-parallel contact region and its three compartments (left side, central part, and right side) of the lamin 300 fragment are indicated by double-headed arrows. Four chains (chains A–D) are in green, blue, violet, and yellow, respectively. Left and right-side parts of the lamin 300 fragment and vimentin are marked as dotted boxes. The N and C terminus of each chain is indicated by N and C, respectively. The contact interfaces between chain A:B and C:D are marked as black boxes labelled with a–e, which are enlarged in **b**. **b** Residues involved in hydrophobic or molecular contacts are shown in the stick representations. See also Supplemental Fig. 4 for interfaces of the vimentin. **c** SEC profile of the lamin 300 fragment A146C mutant and molar ratio of disulfide-bonded proteins in each peak on Superdex 200 10/300 column. Five labelled fractions (left) were analysed using non-reducing SDS-PAGE (middle), and the relative portion of disulfide-bonded protein (tetramer) to free Cys protein (dimer) was calculated based on the band intensity using ImageJ (right). **d** SEC-MALS analysis of lamin 300 fragment (2.5 mg/ml, wild type; left and A146C mutant; middle) in the absence (black line) or presence of a reducing agent TCEP (red line). Relative refractive index (right y-axis) and molecular weight of each peak (left y-axis) are plotted against the elution volume. The protein samples were also analysed using SDS-PAGE in the presence or absence of the reducing agent TCEP (right). Molecular sizes (kDa) of the marker proteins (M) are labelled on the left. Note that MALS might not give the accurate values when two peaks were not resolved. Considering the SDS-PAGE analysis of the fractions (in **c**), the two peaks (middle) correspond to the tetramer and dimer, respectively

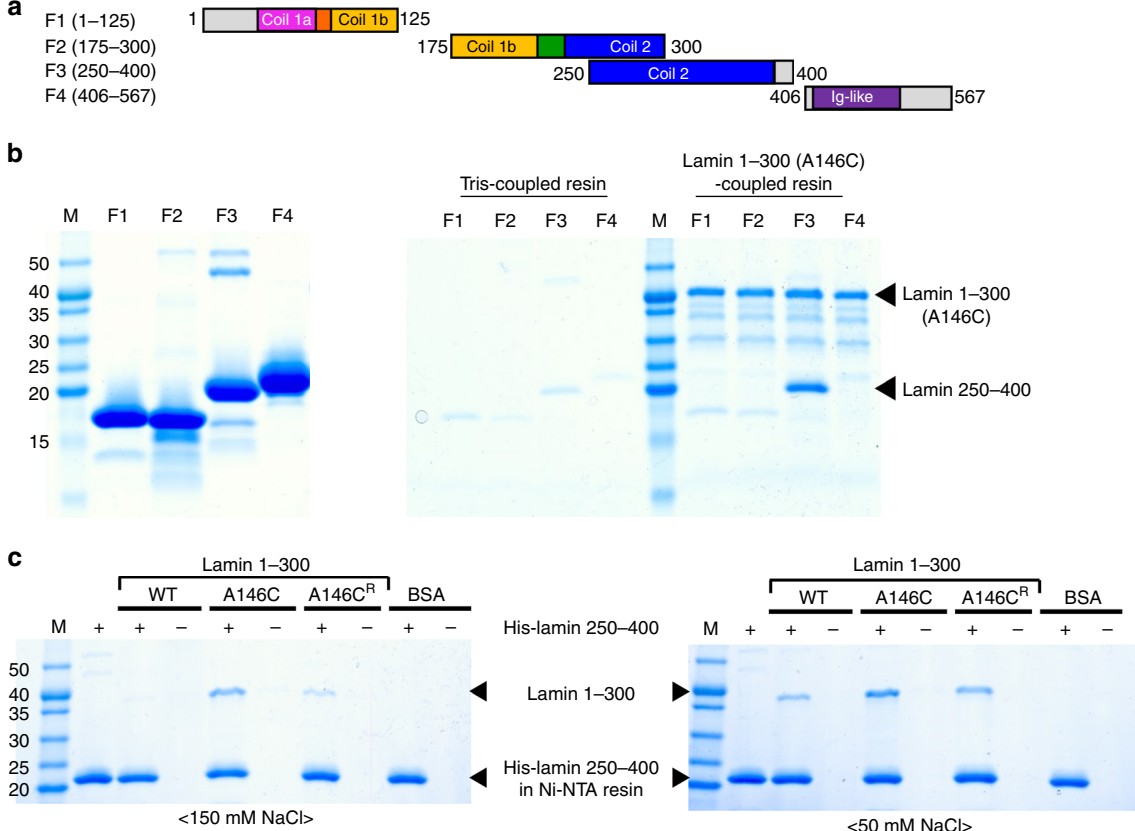

**Fig. 4** Direct binding between the lamin fragments depending on the A11 interaction. **a** Lamin fragments (F1–F4) were used for the binding assay. The sub-domains are coloured differently as shown in Fig. 1a. The start and end residues of the four fragments are indicated. **b** Identification of the binding partners of the lamin 300 fragment. Four purified fragments are shown in the left SDS-PAGE gel. An in vitro pull-down assay was conducted using the immobilized lamin 300 A146C mutant on CNBr-activated resin. Four candidate fragments (66 μM) were incubated with lamin 300 A146C-coupled resin or Tris-coupled resin under the non-reducing condition (right SDS-PAGE gel). **c** Comparison of binding affinity between the lamin 300 variants and F3 depending on A11 interaction. An in vitro pull-down assay was conducted using His-F3 (residues 250–400)-bound Ni-NTA resin. BSA and wild type (WT; dimer), non-reduced A146C (A146C; tetramer), and reduced A146C (A146C^R; dimer) of the lamin 300 fragment were incubated on the empty (−) or His-F3-bound (+) Ni-NTA resins. The resins were pre-equilibrated and washed with the 20 mM Tris-HCl (pH 8.0) buffer containing 150 mM (left) or 50 mM NaCl (right). For A146C^R, the buffer was supplemented with 2 mM TCEP (reducing agent). The bound proteins in the resins were analysed using SDS-PAGE. Molecular weights (kDa) of the marker (M) are labelled on the left

The Y45C mutation in the lamin 300 fragment abolished binding to the C-terminal part of coil 2 (Fig. 8a and Supplementary Fig. 6b). In contrast, we observed that the L59R mutation increased the eA22 interaction in the high salt buffer (Fig. 8a right). Furthermore, isothermal titration calorimetry (ITC) results showed ~40-fold higher binding affinity of L59R mutation than the wild type against the C-terminal part of coil 2 in the

phosphate-buffered saline, based on $K_D$ values (Supplementary Fig. 7). Consistent results were obtained in the nuclear shapes and the distribution of the lamin A/C when the mutant lamin genes were overexpressed in the cell lines (HT1080 and SHSY5Y). Lamin A-Y45C formed a blebbing in the nucleus, and they diffused to the cytosol when mechanical stress was applied to the nucleus by cleaving the chromosomal DNA with a nuclease

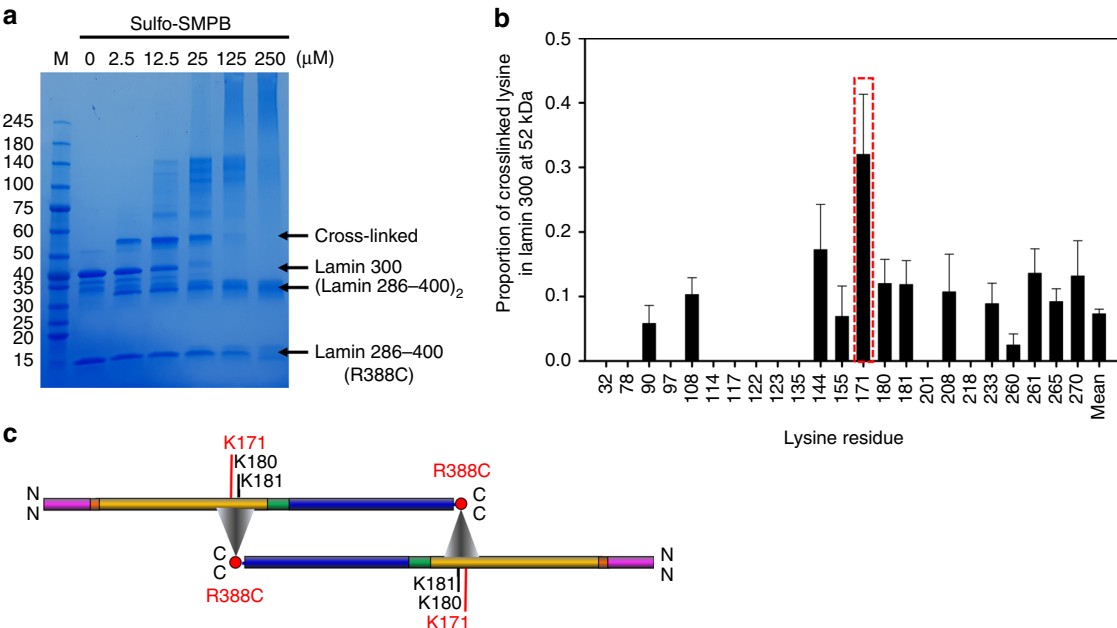

**Fig. 5** The eA22 interaction mapping by MS/MS analysis with cross-linked proteins. **a** The chemical cross-linking assay with the lamin 300 fragment and lamin 286–400 R388C mutant in SDS-PAGE. Protein bands of the lamin 300 fragment and the cross-linked sample are labelled. To block the unreacted chemical cross-linker, excess cysteine was added. The major bands of 'cross-linked' (52 kDa), lamin 300 (38 kDa), dimerized lamin 286–400 R388C [(lamin 286–400)$_2$; 30 kDa], and lamin 286–400 R388C (14 kDa) were excised and subjected to the subsequent MS/MS analysis (Supplementary Data 1). **b** Proportion (p̂) of the Cys388-bound lysine residues at each position was determined by dividing the number of Cys388-bound lysine residues by the total number of observed lysine residues in the pooled data of cross-linked bands. Position of the most probable Lys171 to be cross-linked to Cys388 is shown in a red box in the plot. The error bars indicate each standard error of proportion, which were calculated by $\sqrt{\frac{\hat{p}*(1-\hat{p})}{\text{number of observed peptides}}}$. See also Supplementary Table 2. **c** Schematic representation of the central rod domains of two coiled-coil dimers in the colour code of Fig. 1a, displaying the eA22 interaction based on the results of the MS/MS analysis. Triangles indicate that Arg388 (or Cys388) of coil 2 is around Lys171 (or its neighboring Lys180 or Lys181) of coil 1b in lamin 300 fragment

(benzonase). When lamin A-L59R was overexpressed, stronger lamin aggregates were found at the peripheral region of the nuclear envelopes, probably by reinforced filament formation (Fig. 8b and Supplementary Figs. 8 and 9).

To explain the opposite effects of the Y45C mutation for muscular dystrophy and the L59R mutation for dilated cardiomyopathy, we noted the locations of the two residues. Both Tyr45 and Leu59 residues are in the d position in the heptad repeat (Fig. 2a). The crystal structure further revealed that Tyr45 makes an awkward interaction (with Ile46 at the e position) and Leu59 is ideally positioned in the coiled-coil (Fig. 8c, d). The Y45C mutation is likely to stabilize the coiled-coil interaction in coil 1a with the small and hydrophobic residue by relieving the awkward interaction. This appears to inhibit the interaction with the coil 2. However, the potential disulfide bond formation between the cysteine residues might not be important in the Y45C mutant (Supplementary Fig. 6b). In contrast, the L59R mutation would destabilize the coiled-coil interaction through the non-homologous change, which would augment the interaction with the coil 2 region as shown in Fig. 8a. The reinforced filament formation would also deteriorate dynamic remodeling or correct mesh formation in cells for the robust nuclear envelope. Thus, our findings suggest that the stronger or weaker interactions may be responsible for pathological states, since both cases could cause adverse effects in the formation of robust nuclear structures.

## Discussion
There are many different models for lamin and vimentin, some of which are based on the EM images of refolded proteins or their paracrystalline forms at a low resolution[13,18,24,41,42]. However, to

understand the physiological structure of IF proteins, we now need to focus on the structures representing the native state and high-resolution structures. In this study, we determined the crystal structure of a fragment of lamin containing all the linker segments, directly displaying the A11 interaction between two coiled-coil dimers. This structure was long enough to extend and to connect the previous lamin structures determined by Strelkov et al.[18,26,28], visualizing most structural features of the full-length lamin. A cross-linking study with MS/MS analysis further discovered an interaction, called an eA22 interaction, which acts presumably synergistically with the A11 interaction. By combining the two interactions, we built a complete assembly model at a molecular level, which agrees well with the in situ cryo-ET structure representing the native lamin filament recently provided by Turgay el al.[16].

Typical IF proteins, such as vimentin, form thicker filament structures than lamins[43,44]. The α-helical rod domain of vimentin has a 42 residue-long gap sequence corresponding to the central compartment of coil 1b of lamin, with substantial length differences in the N- and C-terminal unstructured regions (Fig. 3a and Supplementary Figs. 3 and 4). The central rod domain of vimentin is shorter by ~6 nm than that of lamin due to the gap 42 residues. However, the gap 42 residues seemed not to affect the A11 and eA22 interactions because the central compartment of coil 1b was not directly involved in the filament formation of lamin. Thus, it is likely that the overall structures of the α-helical rod domain of typical IF proteins would be shared with those of lamins[45]. Additional interactions, which are present only in typical IF proteins, but not in lamin, would be required to form the final 10-nm-thick filament, which consists of four to six lamin-like filaments with a shortened interval.

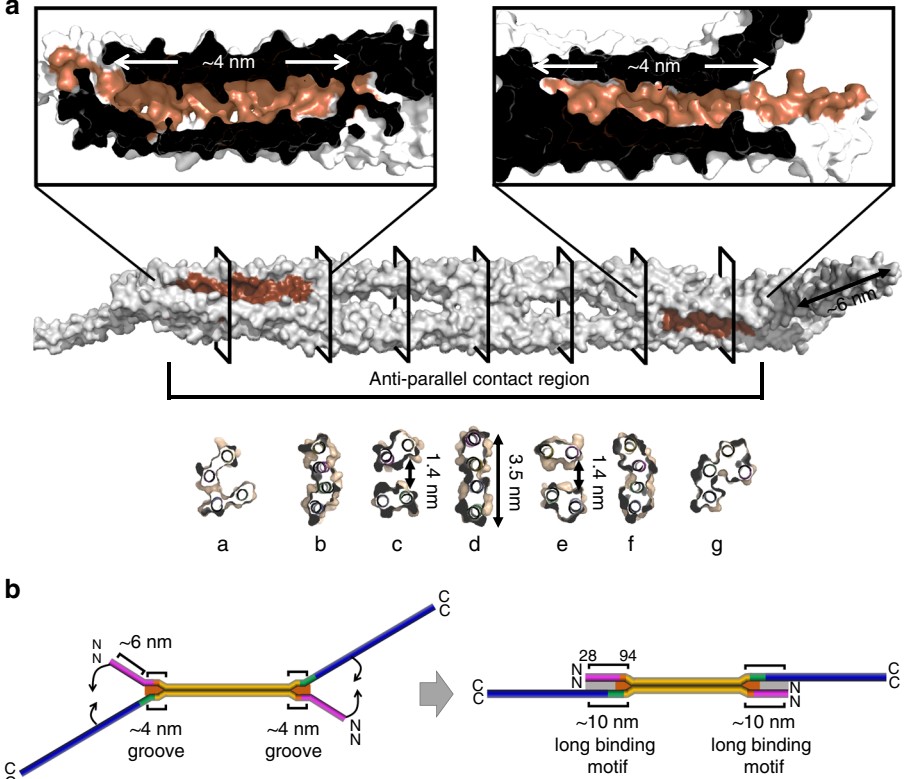

**Fig. 6** The grooves and the space in the lamin A11 tetramer. **a** The lamin tetramer is shown in the surface representation (grey), and the anti-parallel contact region is marked. Two ~4-nm-long grooves are in brown and enlarged in the top panel. The cross-sections, marked with a–g, are shown in the bottom panel. **b** The bent model of the full-length rod domain reflecting the crystal structure of the lamin A11 tetramer is shown in the left panel and the straightened model is shown in the right panel. The sub-domains are coloured differently as shown in Fig. 1a. The ~4 nm grooves and ~6-nm-long spaces are in brown and light grey, respectively. Detailed information for the model construction is depicted in Supplementary Fig. 5

In conclusion, we discovered two essential interactions for lamin assembly by the crystal structures and biochemical studies. These interactions could provide a key to understanding the nuclear function and the assembly mechanisms for IF proteins at the molecular level.

## Methods

**Plasmid construction.** For overexpression of lamin proteins, amplified DNA fragments coding for residues 1–300 (wild-type and A146C mutant), 1–125, 175–300, 250–400, and 406–567 of lamin A/C were inserted into the pProEx-HTa vector (Thermo Fisher Scientific, MA, USA). The used oligonucleotide primer sequences are shown in Supplementary Table 3. The resulting plasmids encoded the His tag and the tobacco etch virus (TEV) protease cleavage site at the N-terminus of the lamin proteins. To prevent overlapping of the peptides from the lamin 300 and 250–400 fragments digested by chymotrypsin for mass spectrometry analysis, we constructed a 286–400 fragment. The arginine 388 residue was replaced with cysteine, which enabled cross-linking of a lysine residue in the lamin 300 fragment through the primary amine and sulfhydryl reactive spacer of sulfo-succinimidyl 4-(N-maleimidophenyl)butyrate (Sulfo-SMPB; Sigma Aldrich, USA). For immunofluorescence staining, amplified DNA fragments encoding wild type and the Y45C and L59R mutants of full-length lamin A/C were inserted into the pcDNA3.1(+) vector (Thermo Fisher Scientific, MA, USA).

**Purification of the recombinant proteins.** To obtain the selenomethionyl-labelled protein for crystallization, the *Escherichia coli* strain B834 (DE3; Novagen, USA) harbouring the plasmid encoding the lamin A/C fragment (residues 1–300) was cultured in M9 medium supplemented with L-(+)-selenomethionine. The protein expression was induced by 0.5 mM IPTG at 30 °C. The cells were harvested by centrifugation and resuspended in lysis buffer containing 20 mM Tris-HCl (pH 8.0) and 150 mM NaCl. The cells were disrupted using sonication and the cell debris was removed using centrifugation. The supernatant was loaded onto Ni-NTA affinity agarose resin (Qiagen, The Netherlands), pre-incubated with lysis buffer. The target protein was eluted with lysis buffer supplemented with 250 mM imidazole. The eluate was treated with TEV protease to cleave the His tag, and then was loaded onto a HiTrap Q column (GE Healthcare, USA). A linear gradient of

increasing NaCl concentration was applied to the HiTrap Q column. The fractions which contained the protein were applied onto a size exclusion chromatography column (HiLoad Superdex 200 26/600 column; GE Healthcare), pre-equilibrated with lysis buffer. The purified protein containing lysis buffer was concentrated to 7 mg/mL and frozen at −70 °C.

For biochemical assays, each plasmid encoding the lamin A/C fragments was transformed into *E. coli* BL21 (DE3; Novagen, USA) and cultured in LB or TB medium. The same procedure was used to purify the proteins. For the His-tagged proteins, the TEV protease treatment was not applied.

**Crystallization and structure determination.** The selenomethionine-labelled lamin protein (7 mg/mL) whose His-tag was cleaved off was crystallized in a precipitation solution containing 0.1 M Tris-HCl (pH 8.5), 0.9 M lithium chloride, and 7% (w/v) PEG 8000 using the hanging-drop vapour diffusion method at 14 °C. Crystals with a size of ~300 μm were used for structure determination. The crystals were dipped in a cryoprotectant solution containing 3.75% (v/v) diethylene glycol, 3.75% (v/v) ethylene glycol, 3.75% (v/v) (±)-2-methyl-2,4-pentanediol, 3.75% (v/v) 1,2-propanediol, 3.75% (v/v) dimethyl sulfoxide, and 3.75 mM 3-(1-pyridino)-1-propane sulfonate and flash-frozen in a nitrogen stream at −173 °C. A single-wavelength anomalous diffraction (SAD) dataset was collected at Pohang Accelerator Laboratory Beamline 5C using Pilatus 3 6M Detector[46] and processed with the HKL2000 package[47,48]. The crystals belong to space group $P2_12_12$ with unit-cell dimensions of $a = 231.3$ Å, $b = 85.0$ Å, and $c = 92.4$ Å. Heavy atom searching was performed using SHELXC/D of the CCP4i package[49,50]. The lamin 300 fragment contained three Met residues in the sequence. However, two Se-Met were identified per protomer in the SAD date by SHELXD. Eight Se-Met sites were identified because the asymmetric unit contained four protomers. Phase calculation and density modification were performed using Phaser of the PHENIX[51,52]. The programmes COOT and PHENIX were used for model building and refinement[52–54]. The final structure was refined at a 3.2 Å resolution ($R_{factor}$ and $R_{free}$ of 20.7% and 26.1%, respectively; Supplementary Table 1)[55].

**SEC-MALS.** Molecular mass was determined with analytical size exclusion chromatography coupled with multi-angle light scattering (SEC-MALS). The protein samples (wild-type and A146C mutant of the lamin 300 fragment; 2.5 mg/ml) were applied to a Superdex 200 Increase 5/150 GL column (GE healthcare), pre-

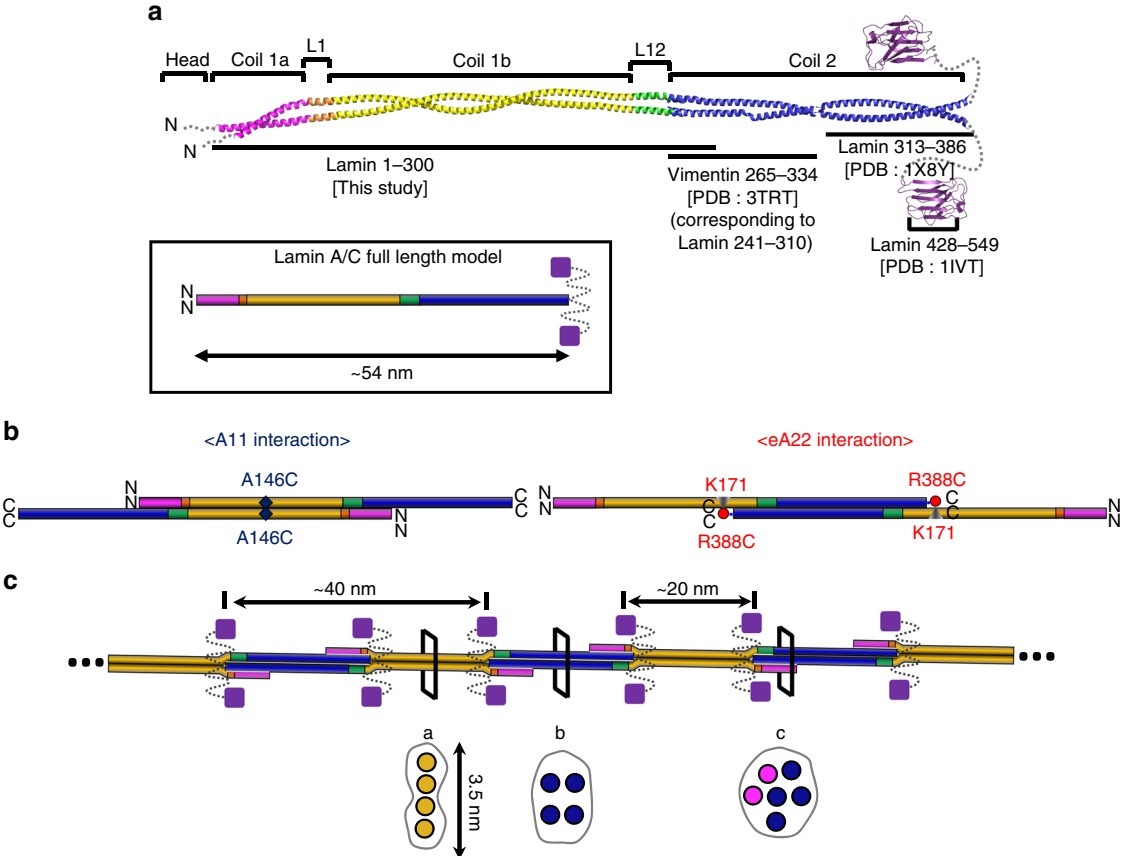

**Fig. 7** Proposed assembly mechanism of lamin by the A11 and eA22 interactions. **a** Coiled-coil dimer model of full-length lamin based on the lamin 300 fragment structure. The C-terminal region of coil 2 (residues 313–386; PDB code: 1 × 8Y[26]) and the Ig-like domain (residues 428–549; PDB code: 1IVT[25]) of human lamin A/C, were added to the structure described in this study. For the N-terminal region of coil 2, the corresponding vimentin fragment (PDB code 3TRT;[42] corresponding to residues 241–310 in lamin A/C) was used. The missing part (residues 283–385) in the central rod domain was built with the typical coiled-coil structure with a rise per residue of 1.5 Å. The unstructured regions (N-terminal head, NLS, and C-terminal tail) are indicated by dotted lines. Schematic drawings of the dimer model of the 54-nm-long full-length lamin A/C is in the box. **b** Two different tetrameric assemblies by the A11 or eA22 interactions. Each band in the same colour code as in Fig. 1a represents the coiled-coil dimer. Left, the A11 interaction-based tetrameric assembly. Right, the eA22 interaction-based tetrameric assembly. The nearby residues mapped by the disulfide or the chemical cross-linking assays are labelled. **c** The model of the mature lamin filament, formed by the successive and alternative A11 and eA22 interactions between the dimers. When a pair of Ig-like domains (violet square) are placed on the both sides of the tetramers, the distance between the neighbouring Ig-like domains is ~20 nm. Schematic cross-sections (a–c) of the lamin filament model are shown in the bottom (circles represent intersection of an α-helix, binding motif are in brown). The cross-section a is equivalent to Fig. 6d, and b to Fig. 6e, while c is an arbitrary arrangement of 6 α-helices

equilibrated with a buffer containing 20 mM Tris-HCl (pH 8.0), 150 mM NaCl, 2 mM tris(2-carboxyethyl) phosphine (TCEP) or a non-reducing buffer containing Tris-HCl (pH 8.0) and 150 mM NaCl. The mass-averaged molecular weights were calculated using Debye analysis with ASTRA 6 software (WYATT, USA).

**Pull-down assays**. For Fig. 4b, we conducted a pull-down assay using immobilized lamin 300 A146C mutant protein. The lamin 300 A146C mutant protein was coupled to CNBr-activated Sepharose resin (GE Healthcare) and/or blocked with excessive 0.1 M Tris-HCl (pH 8.0) buffer containing 150 mM NaCl. Four purified lamin fragments (residues 1–125, 175–300, 250–400, and 406–567; 66 μM) were incubated with the lamin-coupled resin or Tris-blocked resin (for control), which were pre-equilibrated with a 20 mM Tris-HCl (pH 8.0) buffer containing 150 mM NaCl. After washing with the buffer, the resin was analysed using SDS-PAGE.

For Fig. 4c, His-tagged lamin proteins were immobilized on the Ni-NTA resin as bait. BSA or His-tag cleaved lamin proteins as a prey were incubated on the His-lamin immobilized resin pre-equilibrated in a 20 mM Tris-HCl (pH 8.0) buffer containing 150 mM NaCl (or 50 mM NaCl) at room temperature for 30 min. After washing with the buffer supplemented with 20 mM imidazole, the resin was analysed using SDS-PAGE.

**Cross-linking reaction and MS/MS analysis**. Purified lamin 286–400 R388C protein was incubated with 2 mM DTT and changed to a 20 mM 4-(2-hydroxyethyl)-1-piperazineethanesulfonic acid (HEPES) (pH 7.5) buffer containing 50 mM NaCl, and 1 mM EDTA using a HiPrep 26/10 desalting column (GE Healthcare). Sulfo-SMPB (100 mM stock solution) was added in 0, 2.5, 25, 125, and

250 μM to a mixture containing 20 μM lamin 300 and coil 2 (286–400) R388C proteins for 1 h, and the reactions were stopped by a solution containing 40 mM Tris-HCl pH 7.5 and 40 mM L-cysteine. The reaction mixture was subjected to SDS-PAGE, and the protein bands of coil 2 R388C (14 kDa), lamin 300 (38 kDa), and their cross-linked structure (52 kDa) were further analysed by the MS/MS spectrometry.

The corresponding protein bands were excised from the SDS-polyacrylamide gel and were digested in gel by chymotrypsin (Promega, USA) with a low specificity towards the C-termini of tyrosine, phenylalanine, tryptophan, and leucine according to the manufacturer's manual. The digested peptides were extracted from the gel slices, as previously described[48], and analysed on a nLC-Velos Pro mass instrument equipped with a PicoFrit™ column, 100 mm, packed with 5 μm Biobasic® C18 and an EASY-Column™, 2 cm, packed with 5 μm C18 (Thermo Fisher Scientific). The LC condition operated at 0.3 μL min⁻¹ was a 45-min linear gradient from 5 to 40% ACN in a 0.1% formic acid buffer solution, followed by a 10 min column wash with 80% ACN and a 20 min reequilibration to the initial buffer condition. Full mass (MS1) scan was performed in range of m/z 300–2000 in a positive ion mode. Data-dependent MS2 scans of the seven most intense ions were performed from the full scan with scan options of 1.5 m/z isolation width, 25% normalized collision energy, and 30 s dynamic exclusion duration. The acquisitioned MS2 data were primarily analysed by a SEQUEST search with a maximum miscleavage of 2, precursor mass tolerance 1.5 Da, fragment mass tolerance 1.0 Da, dynamic modification of lysine either blocked with a cysteine-conjugate cross-linker (Cys~K, +m/z 362.4) or cross-linked with a Cys388-Leu389 dipeptide (CL~K, +m/z 475.5) of the coil 2 R388C fragment, and static modification of the Cys388 residue blocked with a Tris-conjugate cross-linker

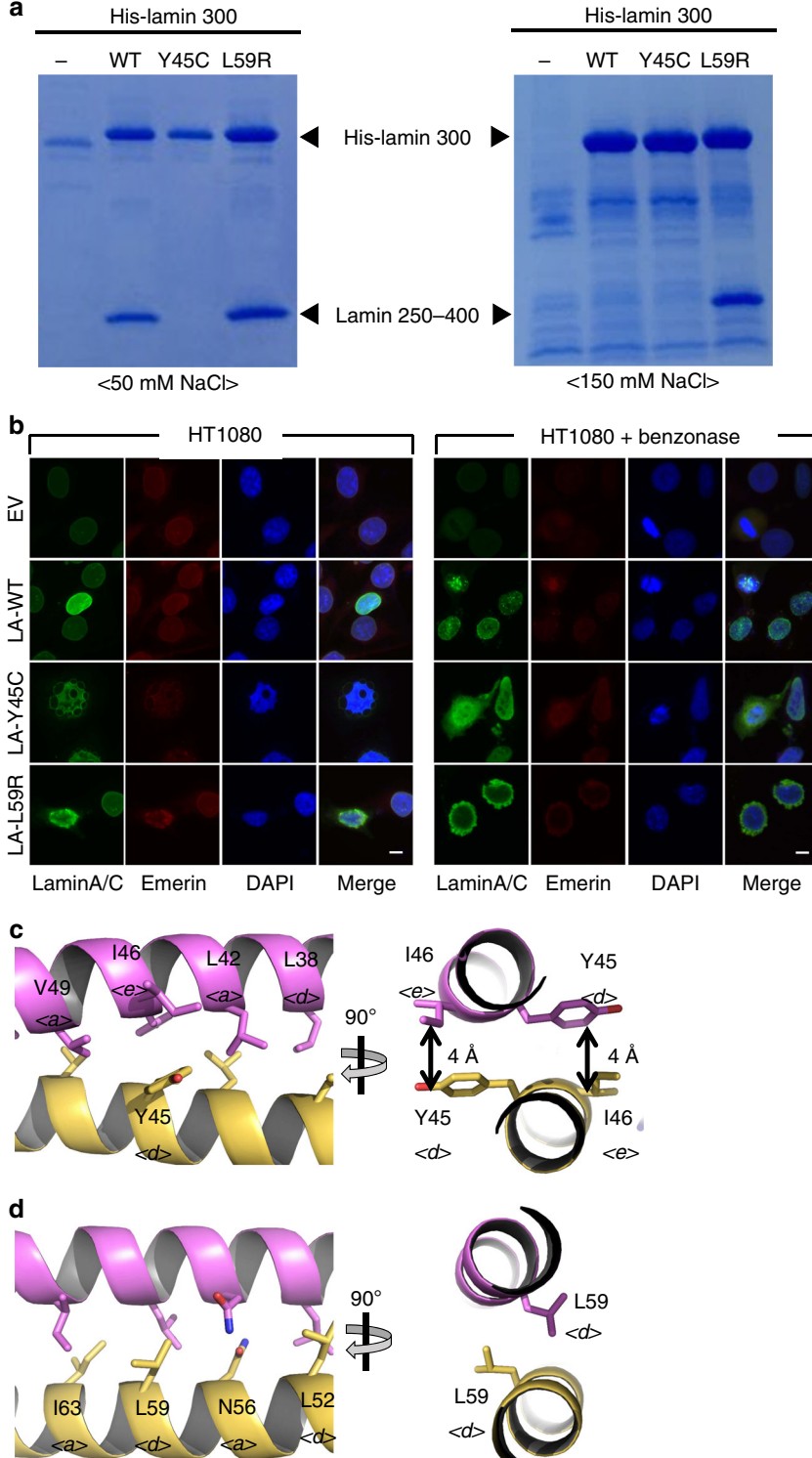

**Fig. 8** The laminopathy-related mutations. **a** Altered eA22 interaction by mutations Y45C or L59R, revealed by in vitro binding assay. Wild-type, Y45C, or L59R mutant proteins of the His-tagged lamin 300 fragment were immobilized on Ni-NTA resin. Lamin 250–400 fragments were incubated on the empty (−) or His-lamin-bound (+) Ni-NTA resins. The resins were pre-equilibrated and washed with the 20 mM Tris-HCl (pH 8.0) buffer containing 50 mM (left) or 150 mM NaCl (right). After washing, the bound proteins were analysed using SDS-PAGE. **b** Nuclear shapes and distribution of lamin A/C Y45C and L59R mutant proteins. Nuclear morphology was examined using fluorescence confocal microscopy after transfection of empty vector (EV), wild-type, or Y45C or L59R mutant of lamin A/C (*LA-WT*, *LA-Y45C*, and *LA-L59R*) into HT1080 cells with or without treatment of 2.5 U benzonase for 5 min. For visualization of the nuclear membrane, cells were stained with lamin A/C (green), emerin (red) and DAPI for DNA (blue). See also Supplementary Figs. 7 and 8. **c**, **d** Structural environment around Tyr45 and Leu59. The orthogonal view of the coiled-coil structure around **c** Tyr45 and **d** Leu59 residues are represented. Each Tyr45 residue interacts with Ile46 residue in another protomer (double-headed arrows). Leu59 residues adopt ideal inter coiled-coil interaction. 'a', 'd', and 'e' positions of the residues in the heptad repeat are labelled with italic letter

(Tris~C, *m/z* 362.4) in the coil 2 fragment band (14 kDa). After filtering out the peptide-to-spectrum matches (PSM) with a peptide probability above 90%, the tandem mass spectra of the cross-linked peptides were manually assigned to the fragment ions generated from the collision-induced dissociation of the precursor ion. We report the results of the PSM in Supplementary Data 1. The manually assigned tandem mass spectra of Tris-crosslinker conjugate Cys388-Leu389 dipeptide and target Lys171 cross-linked with the Cys388-Leu389 dipeptide are shown with extracted ion chromatograms in Supplementary Figs. 10 and 11.

**Isothermal titration calorimetry**. ITC experiments were carried out using an Auto-iTC200 Microcalorimeter (GE healthcare) at Korea Basic Science Institute (Ochang, Korea). His-tagged wild-type or L59R mutant of lamin 300 fragment (20 μM; 0.7 mg/ml) was prepared in the sample cell (370 μl) and his-tagged 250–400 fragment (160 μM; 3 mg/ml) was loaded into the injectable syringe. All samples were prepared in PBS. Titration measurements of 19 injections (2 μL) with 150 s spacing were performed at 25 °C while the syringe was stirred at 750 rpm. The data were analyzed using the MicroCal Origin™ software.

**Immunofluorescence staining**. A human fibrosarcoma cell line (HT1080) and human neuroblastoma cell line (SH-SY-5Y) obtained from ATCC were maintained in liquid medium (DMEM) containing 10% (v/v) FBS, 1% (v/v) antibiotics at 37 °C. HT1080 or SH-SY-5Y cells were seeded on a cover glass and transfected with the plasmid coding wild type, Y45C or L59R mutant of full-length lamin A/C using jetPEI (Polyplus Transfection). After fixing with 1% (w/v) paraformaldehyde (PFA) for 1 h at 4 °C, cells were permeabilized with 0.1% (v/v) Triton X-100 including 2.5 U/ml benzonase (Calbiochem; 71206–3) or mock for 5 min and incubated with a blocking buffer containing PBS and anti-human antibodies (1:400) for 1 h. After washing with PBS twice, the cells were incubated with anti-lamin A/C (sc-376248; Santa Cruz Biotechnology) and anti-emerin (sc-15378; Santa Cruz Biotechnology) primary antibodies (1:200) in the blocking buffer overnight, followed by secondary antibodies (anti-mouse Ab-FITC and anti-rabbit Ab-rhodamine; 1:400) in the blocking buffer for 7 h and mounted. The nucleus was stained with DAPI. The immunofluorescence signal was detected using fluorescence microscopy (Logos).

**Reporting summary**. Further information on research design is available in the Nature Research Reporting Summary linked to this article.

## Data availability

Coordinates and structure factors have been deposited in the Protein Data Bank under accession code 6JLB. The mass spectrometry data have been deposited at the ProteomeXchange Consortium via the PRIDE[56] partner repository with the dataset identifier PXD013144 and PXD01429. The source data underlying Figs. 3c, d, 4b, c, 5a, b, and 8a and Supplementary Figs. 6b and 9a are provided as a Source Data file. Other data are available from the corresponding author upon reasonable request.

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

## Acknowledgements

We would like to thank the Pohang Accelerator Laboratory (Pohang, Republic of Korea) for use of its Beamline 5C equipment, and the ITC facility at the Korea Basic Science Institute (Ochang, Republic of Korea). This research was supported by the Korea Institute of Planning and Evaluation for Technology in Food, Agriculture, and Forestry (IPET; 710012-03-1-HD120; ARC program to NCH), funded by Ministry of Agriculture, Food and Rural Affairs. This study was also supported by grants from the National Research Foundation of Korea (NRF-2017R1A2B2003992 and NRF-2017M3A9R6029755 to NCH, and NRF-2016R1A2B2014493 to YHK). This work was also supported by the BK21 Plus Program of the Department of Agricultural Bio-technology, Seoul National University, Seoul, Korea.

## Author contributions

J.A., I.J., Y.H.K. and S.M.K. performed the experiments. J.A., I.J. and N.C.H. wrote the manuscript. B.J.P. designed the experiments. J.A., I.J., Y.H.K. and N.C.H. interpreted the results. J.A., S.H., S.K. and S.J. provided the protein samples.
