## [Peer Review File · Nature Communications]

Reviewers' comments:

Reviewer #1 (Remarks to the Author):

The authors report the first 3D structure of the N-terminal coiled coil region of lamin A/C, including coil1a, coil1b and part of coil2. Even if it is always complicated to demonstrate the relevance of a 3D structure formed by a coiled coil protein fragment in a crystal, the authors put forward that the diameter of the tetrameric arrangement observed in the crystal is similar to that of the lamin filaments recently observed by CryoET. Moreover, this tetrameric arrangement is also observed in solution if stabilized by an intermolecular disulfide bridge. Finally, the described antiparallel interaction between 2 lamin dimers shows structural similarities with the antiparallel interaction described between 2 vimentin dimers.

The authors also report that the lamin 300 tetramer interacts with lamin fragment from aa 250 to aa 400, mostly corresponding to coil2, and they elegantly show that tetramerization favors this interaction. They propose an inventive model for interaction between the lamin 300 tetramer and the lamin 250-400 fragment. This work is mostly solid, innovative, and is an important step in the field. However, I still have some concerns about the interpretation of the experimental data and very serious concerns about the discussion related to the impact of disease-associated lamin mutations.

- 1) In the introduction, a proper description of the known X-ray structures of lamin A/C coiled coil fragments and proper references to articles reporting the X-ray structures of human lamin A/C fragments are lacking, in particular Dhe-Paganon et al., JBC 2002; Krimm et al., Structure 2002; Strelkov et al., J Mol Biol 2004.
- 2) Details are lacking about the buffer in which lamins were purified (at the 3 steps of the purification: affinity, ion exchange, size exclusion), the buffer in which they were concentrated for crystallogenesis, the amount of proteins and buffer used for the SEC-MALS and pulldown assays reported in Figs. 3 & 4 (ideally it would be in the legends of the Figs, given that it is a critical data for these experiments).
- 3) Writing that "The A11 interaction represents the lamin assembly process" seems to me excessive. The authors nicely demonstrate that, in solution, A146 from one monomer is close to A146 of another monomer, and that formation of a covalent bond between these 2 positions stabilizes a tetrameric structure of the lamin 300 fragment. However, as it is not possible to reconstitute human lamin A/C filaments in vitro, it is complicated to demonstrate that the tetramer is an intermediate in the lamin assembly process. What would be very interesting to demonstrate is whether the tetramer exists within lamin filaments in cells.
- 4) The MS experiment described in Fig. 5 is not very convincing. The graph in B. is difficult to read. It would be clearer to present the grey and black bars on one plot and the red bars on another plot, and to evaluate how significant are the differences between the measurements corresponding to each Lys position, given the error bars. Positions 180 and 181, which are not far from 171, seem to also be reasonable candidates to discuss (given that the cysteine was introduced at the Cter of the lamin 250-400, and is thus probably mobile relative to the rest of the fragment: this also should be discussed). Moreover, does the lamin 250-400 fragment also oligomerize (as observed by SEC-MALS), which could complicate the analysis?
- 5) Figure S5 is important, because it describes the construction of a model for full length lamin A/C, I would keep it as a main figure. I find Fig. 7 not very clear: how are the 14 nm related to the 10 nm displayed on Fig. 6? What is the residue number corresponding to the coil2 position close to the N-terminus of coil1a on Fig 7B? Does this fit with fragment 300 to 400 interacting with the tetramer (because fragment 175-300 doesn't interact but fragment 250-400 does)?
- 6) The sentence "Although we show that the A11 interaction is applied prior to the eA22 interaction during the filament formation ..." seems to me an overinterpretation of the data (see above). The sentence finishes with "We believe that ...", which is more accurate from my point of view.
- 7) Why is the Igfold domain represented as interacting with coil1a or L12, whereas it could also interact with coil1b (given that a 40 aa linker separates residue 388 from the N-ter of the Igfold) in Figure 7? Do you have some experimental data that justify this representation?
- 8) I don't agree AT ALL with the representation of the mutations associated to EDMD shown on

Fig. 8A. First, the description of this figure is confusing because in the main text it is written that “most genetic mutations causing laminopathies were mapped on the interfaces of the A11 and eA22 interactions, suggesting their importance ...” and in the legend of Fig.8A, it is said that only mutations associated to muscular dystrophies were analyzed. As there are plenty of lamin A/C mutations causing diseases other than muscular dystrophies, the two sentences do not mean the same thing. Then, mutations causing muscular dystrophies are distributed within the whole LMNA gene, there is no hotspot, there are plenty of these mutations in the Igfold domain for ex, which is not part of the A11 and eA22 interfaces. I don't see in this manuscript any argument that would justify that there are more mutations causing muscular dystrophies in the lamin A/C regions involved in these interfaces than elsewhere (except that the interfaces contain most of the coiled coil regions). In the absence of such arguments, it is incorrect to state that “most genetic mutations causing laminopathies were mapped on the interfaces of the A11 and eA22 interactions”.

9) Then the authors focus on 2 mutations: they “selected two mutations (Y45C and L59R) that are related to muscular dystrophy (ref. 11)”. I don't see any data on these 2 mutations in ref. 11. Could the authors add accurate references proving that these two mutations are associated with muscular dystrophy? I found a reference for L59R (Nguyen et al., BBRC 2007), however this mutation is described as associated with MANDIBULOACRAL DYSPLASIA AND PROGEROID FEATURES!!!! References to check for Y45C: Bonne et al., Ann. Neurol 2000 & Scharner et al., Hum Mutat 2011. As the authors do not observe the same impact of the 2 mutations on lamin intermolecular interaction, it could be interesting to check the associate diseases, and discuss the experimental data in the frame of these diseases!

10) Fig 8B: it is clear that mutation Y45C abolishes the interaction, however it is not clear that mutation L59R increases the affinity of lamin 300 for the lamin fragment 250-400, additional results based on a complementary experimental technique should be obtained to justify this latter conclusion.

11) Fig. 8C: these results are very interesting, however if the conclusion is that mutation Y45C induces a decrease in lamin A/C level, that the related immunofluorescence data should be quantified.

12) Fig. 8F: it seems that Y45C does not interact with the lamin fragment 250-400 both in the absence and in the presence of TCEP. Moreover, I am not sure that a disulfide bridge can be formed in the reducing conditions found in the nucleus, this should be at least mentioned. In summary, the negative impact of mutation Y45C on lamin 250-400 binding is probably unrelated to the capacity of cysteines to form disulfide bridges. And this figure could be transferred to the Supplementary information section.

13) What are the arguments to say that coil1a has to be destabilized to allow binding to lamin fragment 250-400? This is mentioned several times in the paragraph reporting the impact of mutations Y45C and L59R on lamin fragment 250-400 binding.

14) One important question on these two mutants is maybe first: do the mutated lamin 300 fragments still form tetramers? Can they be crystallized in the same conditions as the WT protein fragment?

15) Discussion: the following paragraph is not correct (see above): “Our assembly model gave the molecular interpretation of the pathological lamin mutations. We found that most pathological mutations are associated with the overlapping zone of the two interactions. Ensuing studies indicated that the augmented affinity, as well as reduced affinity between lamin dimers, interferes with the proper lamin structure and function.”

16) Discussion: if L59R is the studied example of a mutation causing a progeroid disease, given that it does not modify interaction with lamin fragment 250-400, there is no argument to argue that “altered A11 and eA22 interactions might contribute to the normal ageing process” and that this study “gives structural basis for developing novel anti-ageing drugs to prevent the normal ageing in humans”.

In summary, this structural study is exciting, very important for the field, the model of lamin assembly that is suggested is new and interesting. However, the use of disease-causing mutants to study the function and related interactions of lamin 300 is described in a very confusing manner, arguments related to muscular dystrophies are mixed with arguments related to ageing diseases, and thus the discussion is based on several inaccurate arguments. This seriously weakens the manuscript and should be reconsidered.

Reviewer #2 (Remarks to the Author):

This is a well done study in which the crystal structure of the human lamin fragment (1-300) in a tetrameric form was determined. Structural analysis in combination with biochemical studies showed an "A11 interaction" between the two coiled-coil dimers of lamin. Moreover, a new "eA22 interaction" between lamin dimers was further discovered through chemical cross-linking and mass spectrometry analysis. Based on these results, a model was proposed to explain the lamin assembly process.

I have some minor concerns on this study.

1. Based on the analysis of the tetrameric structure of lamin 300, the authors introduced A146C mutation and perform SEC-MALS and SDS-PAGE analysis to conclude (Page 7, line 8-10) that "We first observed that the A146C variant protein of the lamin 300 fragment formed a disulfide bond in SDS-PAGE, suggesting that the tetrameric arrangement is formed in solution (Figs. 3C and D)". However, more evidences are required to prove that the tetramer formed by mutant A146C is identical to the tetramer observed in the crystal structure.
2. Have the authors tried to crystallize the A146C tetramer, since it is much more stable than the wild-type tetramer?
3. To compare the binding affinity of F3 fragment with wild type and mutant lamin 300, the binding between these proteins should be measured quantitatively by the methods of ITC or SPR.
4. Additionally, SEC-MALS did not give accurate molecular weight when the two peaks are overlapped (Figure 3D, the middle pannel). Other methods like Analytical Ultracentrifugation or Mass Spectrometry might be applied to confirm the molecular weight of A146C mutant.

Reviewer #3 (Remarks to the Author):

This is a very important contribution to our understanding of lamin structure and assembly. To me it squares the circle for the vertebrate and invertebrate intermediate filament proteins that have the "lamin 42 residue extension" by bringing the A-type lamins into the same orbit as their cytoplasmic cousins. Antiparallel utilizing staggered coiled coils is good. Pivotal questions remain (eg selectivity cytoplasmic versus nuclear for the longer helices), but these data will focus minds and activity. Moreover, this is a HUGE body of work that has been compiled to reach this point and the authors/teams should be congratulated for their contribution.

The structures are excellent given the material under investigation. I make a few comments below after consulting an experienced crystallographer where additional explanatory text/citations could be included to evidence the high quality of the data. There are also some comments on the interpretation of the cross-linking data. In particular, the authors should be mindful of the fact that the creation of fragments has been a necessity to deliver structural information, but is also not the filament state. Coiled coils represent minima within the energy landscapes of these fragments, but tweak the system by constraining a region (eg Cys388 or Cys146 and this could influence that landscape. Indeed that is borne out by the interaction of lamin fragments with the red/ox C146 mutant and this argument is used in the Discussion to explain the potential impact of disease-causing mutations with the LMNA proteins. So I agree with the authors on this point in the Discussion, but would encourage them to help the community by alerting us to some of the remaining questions.

The interpretation of the fragment binding and cross-linking data to evidence the A11 and A22 interactions are solid, although with the cross-linking data there are a few points for clarification listed below that I would appreciate the authors considering. Particularly it is worth remembering that cross-linkers are not zero length - in fact 1.1nm for sulpho-SMBP. By virtue of the combined aliphatic/aromatic linker the cross-links are not constrained in their chemical reactivity meaning that the lysine covalently bound represents a snapshot within a spectrum of potential candidates, plus the fact that cross-linkers will drive fragment association toward an assembly end point (Fig 3; SEC-MALS dimer-tetramer mix - not all tetramers; cf Fig. 5), which could potentially differ to the native lamin filament. There is sufficient evidence within the submission that the authors are

mindful of such possibilities.

In conclusion, this is a very important contribution. The authors might consider some textual changes in answer to the points below.

Points to consider:

The crystal structure is complex so it would be helpful to include the crystal size and type of detector (with appropriate citation).

Then to respect the efforts of other crystallographers, would the authors consider including the following citation given the utility of the program SHELX/C/D/E? (Sheldrick GM, Acta Cryst (2010) D66, 479-485 or Sheldrick GM Acta Crystals (2008) A64, 112-122).

Since the side chain assignment was challenging at 3.2 Angstrom, would the authors state how many Se-Met there were per promoter and how many were identified in the SAD data by ShelxD?

Returning the acknowledgement of other crystallographers, the use of Rfree should perhaps include the Brunger citation: Brunger, Nature(1992) 355:472-475.

Likewise, if ncs restraints were used, as is likely, then citing the appropriate literature would be good: Knowledge-based (Jead, JJ et al. Acta Cryst (2012) D68, 381-390) or Local (Uson, I et al Acta Cryst (1999) D55, 1158-67) should be cited.

The interpretation of the SEC-MALS data in Figure 3 could benefit from some expansion. The C146 construct is clearly in dynamic equilibrium between dimer and tetramer states - evidencing the coiled-coil interaction is fluid. Also with reference to Figure 5 the purity of this preparation seems far superior to that used in Figure 5, which then un-necessarily complicates the interpretation in that figure.

The interpretation of the data in Figure 5 could be improved. The gel data in panel A indicate that either proteolytic degradation has occurred or that the cross-linker induces faster migrating species due to intra-chain cross-links. The fact the gel is not properly destained doesn't help. It would help the reader to have these bands in the 35kDa region explained.

The data here also evidence the different conformational states captured after cross-linking. See the bands in the region 100-140kDa. A comment about any proteomic/cross-linking analyses would be helpful as gel mobility is the only identifier for crosslinks between lamin 300/lamin 286-400 or between themselves.

Figure 5B would benefit from more information. The right hand Y-axis is disproportionate to the effect and the interpretation preferred from these data is that K171 is the target Lysine. To me either 180 or 181 are just as good as Lys171. The cross linker has a 1.1nm reach and perhaps the surface geometry of the coiled coils would allow these to be just as involved in this A22 interaction.

Then the authors might want to embellish the discussion concerning the role of the 42-residue lamin rod feature and how this might impact higher-order assemblies. They might like to consider that the literature contains an example of acytoplasmic IF with just such a feature that appears to incorporate into filaments lacking the rod extension (Gene. 1998 Apr 28;211(1):19-27). It would be helpful to the field to have some speculation here to guide future investigations.

Points for Clarification:

An abbreviation list would be helpful - and ensuring that names in full are give at first mention eg Sulfo-SMPB in the text of the manuscript and not hidden in supplementary tables/legends.

Please indicate in the Materials and Method whether the His-tag was removed before crystallisation - it has the TEV site after all.

Would the authors like to define the last shell in Table S1 (3.21- ??? A). This is because the rms deviations are very small which could/would have impacted the refinement.

Reviewer #1 (Remarks to the Author):

Comment 1: In the introduction, a proper description of the known X-ray structures of lamin A/C coiled coil fragments and proper references to articles reporting the X-ray structures of human lamin A/C fragments are lacking, in particular Dhe-Paganon et al., JBC 2002; Krimm et al., Structure 2002; Strelkov et al., J Mol Biol 2004.

Response: We added the references to the revised manuscript as suggested by the reviewer (page 4).

Comment 2: Details are lacking about the buffer in which lamins were purified (at the 3 steps of the purification: affinity, ion exchange, size exclusion), the buffer in which they were concentrated for crystallogenesis, the amount of proteins and buffer used for the SEC-MALS and pulldown assays reported in Figs. 3 & 4 (ideally it would be in the legends of the Figs, given that it is a critical data for these experiments).

Response: We incorporated the experimental conditions in detail in the revised manuscript (See Materials and Methods and figure legends).

Comment 3: Writing that “The A11 interaction represents the lamin assembly process” seems to me excessive. The authors nicely demonstrate that, in solution, A146 from one monomer is close to A146 of another monomer, and that formation of a covalent bond between these 2 positions stabilizes a tetrameric structure of the lamin 300 fragment. However, as it is not possible to reconstitute human lamin A/C filaments in vitro, it is complicate to demonstrate that the tetramer is an intermediate in the lamin assembly process. What would be very interesting to demonstrate is whether the tetramer exists within lamin filaments in cells.

Response: We changed the subheading into “The lamin A11 interaction is important in the lamin assembly process” in the revised manuscript (page 7). As we mentioned in the manuscript, it was established that the dimeric form of lamin A/C is the fundamental unit in the soluble pool (Quinlan, R. A. *et al. J Mol Biol* **192**, 337-349 (1986)). Since several cysteine residues are in 301-C-terminal part of lamin, the A146C mutation could make the

results complicated in the full-length lamin in cells. We believe that further and extensive investigations are required to demonstrate the lamin assembly in cells.

Comment 4: The MS experiment described in Fig. 5 is not very convincing. The graph in B. is difficult to read. It would be clearer to present the grey and black bars on one plot and the red bars on another plot, and to evaluate how significant are the differences between the measurements corresponding to each Lys position, given the error bars. Positions 180 and 181, which are not far from 171, seem to also be reasonable candidates to discuss (given that the cysteine was introduced at the Cter of the lamin 250-400, and is thus probably mobile relatively to the rest of the fragment: this also should be discussed). Moreover, does the lamin 250-400 fragment also oligomerize (as observed by SEC-MALS), which could complicate the analysis?

Response: The Fig. 5A was revised to clearly show distinct bands of lamin 300 (38 kDa), coil 2 (286-400) R388C (14 kDa), a dimer size of coil 2 R388C (30 kDa), and a crosslinked structure of lamin 300 with coil 2 R388C (52 kDa), which were subjected to the MS experiment. In Fig. 5B, the proportion and standard error of the Cys388-bound lysine residues in the lamin 300 fragment were calculated by pooled data from the MS/MS analysis. The standard errors give how significant the differences between the measurements corresponding to each Lys position are. To clarify the result, we removed the Cys-blocked peptides and the ratio to the number of the CL-linked peptide to the number of the Cys-blocked peptides. We believe that the frequency of the cross-linked (or CL-linked) peptides can represent the cross-linking efficiency better than the ratio. We agree with the reviewer's suggestion that positions of Lys180 and Lys181, which are close to the Lys171 site, are likely involved in the eA22 interaction due to the mobile nature of crosslinkers (page 8-9). To address the reviewer's comment, we changed Fig. 5 in the revised manuscript.

Unfortunately, we didn't perform the SEC-MALS with the lamin 250-400 fragment. However, we believe that the lamin 250-400 fragment is dimeric in solution. A previous paper (Kapinos, L. E. *et al.*, *J Mol Biol* **396**, 719-731, doi:10.1016/j.jmb.2009.12.001 (2010)) reported that a recombinant lamin fragment (residues 264-402), which is very similar to our fragment, was dimeric in solution. Moreover, it is likely that most fragments of lamin should

be dimeric or monomeric in solution because the full-length lamin was known as a dimer in solution.

Comment 5: Figure S5 is important, because it describes the construction of a model for full length lamin A/C, I would keep it as a main figure. I find Fig. 7 not very clear: how are the 14 nm related to the 10 nm displayed on Fig. 6? What is the residue number corresponding to the coil2 position close to the N-terminus of coil1a on Fig 7B? Does this fit with fragment 300 to 400 interacting with the tetramer (because fragment 175-300 doesn't interact but fragment 250-400 does)?

Response: We agree with the reviewer's comment. The model for the full length lamin A/C has been moved to the main figure, as suggested by the reviewer (see Figure 7 in the revised manuscript).

We realized that the length indications in Figure 7 seems to be confusing. The 10-binding motif was measured from the crystal structure by adding the lengths of coil 1a and the groove in coil 2. The 14-nm overlapping region was estimated from the chemical cross-linking experiment for binding to the C-terminal part of coil 2. Based on our assembly model between two tetramers, the 14-nm overlapping region includes the 10-nm binding motif. In this revision, we revised the Fig.7 to clarify the relation between the 10-nm binding motif and 14-nm overlapping region and is moved to Supplementary Fig. 5 (page 10; Figure 7).

As the reviewer commented, we believe that the region of 300-400 is important in interaction with the lamin 300 fragment based on the observation that fragment 175-300 did not interact while the fragment 250-400 did. However, the experimental results might not be simple because the oligomeric state of the fragments and the frayed alpha-helical conformation at the end would also affect the protein-protein interactions.

Comment 6: The sentence “Although we show that the A11 interaction is applied prior to the eA22 interaction during the filament formation ...” seems to me an overinterpretation of the data (see above). The sentence finishes with “We believe that ...”, which is more accurate from my point of view.

Response: We also believe that A11 and eA22 interactions occur simultaneously. To prevent misunderstanding, we changed the sentence (See page 10).

“Sequentially or simultaneously, coil 2 of the coiled-coil dimer was inserted to form an A11 tetramer by an eA22 interaction with ~14 nm-long overlapping, including 10-nm-long binding motif, and ~40 nm-long intervals between the C-terminal ends of two coiled-coil dimers in the same direction (Figs. 7b and Supplementary Fig. 5b).”

“We believe that the A11 and eA22 interactions would occur simultaneously and synergistically for the further extensions of the lamin filament (Figs. 7b and c).”

Comment 7: Why is the Igfold domain represented as interacting with coil1a or L12, whereas it could also interact with coil1b (given that a 40 aa linker separates residue 388 from the N-ter of the Igfold) in Figure 7? Do you have some experimental data that justify this representation?

Response: Actually, we did not intend to represent the interaction between Ig-like domain with coil 1a or L12. We changed the Figure 7, depicting the separation between Ig-fold domain and coils.

Comment 8: I don't agree AT ALL with the representation of the mutations associated to EDMD shown on Fig. 8A. First, the description of this figure is confusing because in the main text it is written that “most genetic mutations causing laminopathies were mapped on the interfaces of the A11 and eA22 interactions, suggesting their importance ...” and in the legend of Fig.8A, it is said that only mutations associated to muscular dystrophies were analyzed. As there are plenty of lamin A/C mutations causing diseases other than muscular dystrophies, the two sentences do not mean the same thing. Then, mutations causing muscular dystrophies are distributed within the whole LMNA gene, there is no hotspot, there are plenty of these mutations in the Igfold domain for ex, which is not part of the A11 and eA22 interfaces. I don't see in this manuscript any argument that would justify that there are more mutations causing muscular dystrophies in the lamin A/C regions involved in these interfaces than elsewhere (except that the interfaces contain most of the coiled coil regions).

In the absence of such arguments, it is incorrect to state that “most genetic mutations causing laminopathies were mapped on the interfaces of the A11 and eA22 interactions”.

Response: To address the reviewer’s comment, we changed the sentence, and moved Fig. 8a to the supplemental figure in the revised manuscript (Page 10; Supplementary Fig. 6a).

“We noted that many genetic mutations causing laminopathies were mapped on the interfaces of the A11 and eA22 interactions, suggesting their importance in the formation of a functional nucleus (Supplementary Fig. 6a).”

Comment 9: Then the authors focus on 2 mutations: they “selected two mutations (Y45C and L59R) that are related to muscular dystrophy (ref. 11)”. I don’t see any data on these 2 mutations in ref. 11. Could the authors add accurate references proving that these two mutations are associated with muscular dystrophy? I found a reference for L59R (Nguyen et al., BBRC 2007), however this mutation is described as associated with MANDIBULOACRAL DYSPLASIA AND PROGEROID FEATURES!!!! References to check for Y45C: Bonne et al., Ann. Neurol 2000 & Scharner et al., Hum Mutat 2011. As the authors do not observe the same impact of the 2 mutations on lamin intermolecular interaction, it could be interesting to check the associate diseases, and discuss the experimental data in the frame of these diseases!

Response: We added accurate references for two mutations (page 10).

Comment 10: Fig 8B: it is clear that mutation Y45C abolishes the interaction, however it is not clear that mutation L59R increases the affinity of lamin 300 for the lamin fragment 250-400, additional results based on a complementary experimental technique should be obtained to justify this latter conclusion.

Response: To analyze quantitatively the difference of the binding affinity between wild type or L59R mutant of the lamin 300 and coil 2 fragment, we measured the K_D value of them by using ITC. The L59R mutant of lamin 1-300 (K_D of 0.05 μM) was bound to the lamin 250-400 more strongly than the wild type (K_D of 2.1 μM). Moreover, we added the experimental result in which mutation L59R has clearly stronger affinity for lamin 250-400 than wild type

in a high salt buffer (Fig. 8a). These results were added in Supplementary Fig. 7 and Fig. 8a *right* in the revised manuscript. We believe that these results can justify our conclusion.

Supplementary Fig. 7. Isothermal titration calorimetry (ITC) analysis for the binding of lamin 300 fragments (WT; a and L59R; b) to coil 2 fragment (250-400). The top panels represent the raw data plot as a series of peaks corresponding to the heat change ($\mu\text{cal/s}$) resulting from titration of lamin 300 fragment ($20 \mu\text{M}$; $370 \mu\text{L}$) with 19 injections of coil 2 fragment ($160 \mu\text{M}$; $2 \mu\text{L}$ per one injection). The bottom panels show the integrated heat pulses in the top panels.

Figure 8. The laminopathy-related mutations. a Altered eA22 interaction by mutations Y45C or L59R, revealed by *in vitro* binding assay. Wild type, Y45C, or L59R mutant proteins of the His-tagged lamin 300 fragment were immobilized on Ni-NTA resin. Lamin 250-400 fragment were incubated on the empty (-) or His-lamin-bound (+) Ni-NTA resins. The resins were pre-equilibrated and washed with the 20 mM Tris-HCl (pH 8.0) buffer containing 50 mM (**left**) or 150 mM NaCl (**right**). After washing, the bound proteins were analysed using SDS-PAGE.

Comment 11: Fig. 8C: these results are very interesting, however if the conclusion is that mutation Y45C induces a decrease in lamin A/C level, that the related immunofluorescence data should be quantified.

Response: We made a mistake. Morphological changes were found in the nucleus of lamin Y45C mutant cells, but not the decrease in the level of lamin. However, this does not change our conclusion. We measured the expression level of lamin (wild type and mutants) by western blotting. As you can see, the expression level of lamin was not changed in the cells expressing the mutant lamins. We corrected the description in the revised manuscript. Thank you for providing us the opportunity to correct it (page 11).

Comment 12: Fig. 8F: it seems that Y45C does not interact with the lamin fragment 250-400 both in the absence and in the presence of TCEP. Moreover, I am not sure that a disulfide bridge can be formed in the reducing conditions found in the nucleus, this should be at least mentioned. In summary, the negative impact of mutation Y45C on lamin 250-400 binding is probably unrelated to the capacity of cysteines to form disulfide bridges. And this figure could be transferred to the Supplementary information section.

Response: This is another mistake. We think that we were a little bit rushed in preparing the manuscript. We corrected the description on Y45C. As the reviewer commented, the disulfide bond formation in Y45C mutant proteins were not important for explaining the decreased binding affinity to the lamin 250-400 fragment. Fig. 8F was transferred to the Supplementary Fig. 6b as suggested by reviewer (page 11; Supplementary Fig. 6b).

“The Y45C mutation is likely to stabilize the coiled-coil interaction in coil 1a with the small and hydrophobic residue by relieving the awkward interaction. This appears to inhibit the interaction with the coil 2. However, the potential disulfide bond formation between the cysteine residues might not be important in the Y45C mutant (Supplementary Fig. 6b).”

Comment 13: What are the arguments to say that coil 1a has to be destabilized to allow binding to lamin fragment 250-400? This is mentioned several times in the paragraph reporting the impact of mutations Y45C and L59R on lamin fragment 250-400 binding.

Response: There are several experimental data indicating on the possibility of instability of coil 1a in the vimentin (Meier, M. et al. J Mol Biol 390, 245-261 (2009))

doi:10.1016/j.jmb.2009.04.067). In addition, it seems to be important that rearrangement of coil 1a to bind to the 250-400 fragment in our study. Thus we believe that the rearrangement of the coil1a might be required to make the interaction for elongation. In fact, we are preparing another manuscript focusing on these observations.

Comment 14: One important question on these two mutants is maybe first: do the mutated lamin 300 fragments still form tetramers? Can they be crystallized in the same conditions as the WT protein fragment?

Response: We failed to obtain the crystals of the mutant proteins. Based on the SEC result, we didn't see the difference from the wild type fragment in solution. Thus we believe that the mutant fragments are dimeric as the wild type fragment.

Comment 15: Discussion: the following paragraph is not correct (see above): “Our assembly model gave the molecular interpretation of the pathological lamin mutations. We found that most pathological mutations are associated with the overlapping zone of the two interactions. Ensuing studies indicated that the augmented affinity, as well as reduced affinity between lamin dimers, interferes with the proper lamin structure and function.”

Response: To address the reviewer's comment, we deleted the paragraph.

Comment 16: Discussion: if L59R is the studied example of a mutation causing a progeroid disease, given that it does not modify interaction with lamin fragment 250-400, there is no argument to argue that “altered A11 and eA22 interactions might contribute to the normal ageing process” and that this study “gives structural basis for developing novel anti-ageing drugs to prevent the normal ageing in humans”.

Response: We deleted the sentences regarding the ageing.

Comment 17: In summary, this structural study is exciting, very important for the field, the model of lamin assembly that is suggested is new and interesting. However, the use of disease-causing mutants to study the function and related interactions of lamin 300 is described in a very confusing manner, arguments related to muscular dystrophies are mixed

with arguments related to ageing diseases, and thus the discussion is based on several inaccurate arguments. This seriously weakens the manuscript and should be reconsidered.

Response: We appreciate the reviewer's interest in our model of lamin assembly. We revised the manuscript to reflect most of the reviewer's comments. In particular, the manuscript was revised to alleviate the reviewer's concerns about muscular dystrophies and aging.

Reviewer #2 (Remarks to the Author):

Comments 1 and 2: Based on the analysis of the tetrameric structure of lamin 300, the authors introduced A146C mutation and perform SEC-MALS and SDS-PAGE analysis to conclude (Page 7, line 8-10) that “We first observed that the A146C variant protein of the lamin 300 fragment formed a disulfide bond in SDS-PAGE, suggesting that the tetrameric arrangement is formed in solution (Figs. 3C and D).”. However, more evidences are required to prove that the tetramer formed by mutant A146C is identical to the tetramer observed in the crystal structure. Have the authors tried to crystallize the A146C tetramer, since it is much more stable than the wild-type tetramer?

Response: We think that more evidences are required to prove that the A11 tetramer formed by mutant A146C. However, we believe that what we presented in this study would be the best biochemical evidences. Without actual structure of A146C, all the biochemical evidence might be not enough to prove the tetrameric arrangement of the A146C mutant. In fact, we have attempted to determine the crystal structure of the lamin 300 A146C mutant. We obtained the crystals of A146C under the same crystallization conditions as the reviewer (and we) expected, which strongly indicated that the A146C shows the same tetrameric arrangement. Unfortunately, the crystals were too small to proceed the structural works. We are now under way to improve the crystallization conditions. We hope that we solve the structure at better resolution in the near future.

Lamin 1-300 WT

Lamin 1-300 S146C

Comment 3: To compare the binding affinity of F3 fragment with wild type and mutant

lamin 300, the binding between these proteins should be measured quantitatively by the methods of ITC or SPR.

Response: It was obvious that mutation Y45C of lamin 300 fragment abolishes the eA22 interaction considering Fig. 8a and Supplementary Fig. 6b, as mentioned by reviewer 1. To analyze quantitatively the difference of the binding affinity between wild type or L59R mutant of the lamin 300 fragment and coil 2 fragment, we measured the K_D values of them by using ITC. The L59R mutant of lamin 1-300 (K_D of 0.05 μM) was bound to the lamin 250-400 more strongly than the wild type (K_D of 2.1 μM). Moreover, we added the experimental result in which mutation L59R has clearly stronger affinity for lamin 250-400 than wild type in a high salt buffer (Fig. 8a). These results were added in Supplementary Fig. 7 and Fig. 8a *right* in the revised manuscript. We believe that these results can justify our conclusion.

Supplementary Fig. 7. Isothermal titration calorimetry (ITC) analysis for the binding of lamin 300 fragments (WT; a and L59R; b) to coil 2 fragment (250-400). The top panels represent the raw data plot as a series of peaks corresponding to the heat change ($\mu\text{cal/s}$) resulting from titration of lamin 300 fragment (20 μM ; 370 μL) with 19 injections of coil 2 fragment (160 μM ; 2 μL per one injection). The bottom panels show the integrated heat pulses in the top panels.

Figure 8. The laminopathy-related mutations. a Altered eA22 interaction by mutations Y45C or L59R, revealed by *in vitro* binding assay. Wild type, Y45C, or L59R mutant proteins of the His-tagged lamin 300 fragment were immobilized on Ni-NTA resin. Lamin 250-400 fragment were incubated on the empty (-) or His-lamin-bound (+) Ni-NTA resins. The resins were pre-equilibrated and washed with the 20 mM Tris-HCl (pH 8.0) buffer containing 50 mM (**left**) or 150 mM NaCl (**right**). After washing, the bound proteins were analysed using SDS-PAGE.

Comment 4: Additionally, SEC-MALS did not give accurate molecular weight when the two peaks are overlapped (Figure 3D, the middle panel). Other methods like Analytical Ultracentrifugation or Mass Spectrometry might be applied to confirm the molecular weight of A146C mutant.

Response: We admit that the molecular weights from SEC-MALS might not be accurate. However, we believe that only the crystal structure of the A146C mutant can prove its oligomeric state. Since we obtained the same crystal forms under the same crystallization condition, as we showed above, it is very likely that the A146C mutant fragment has the same structure as in the wild type one.

Reviewer #3 (Remarks to the Author):

Comment 1: The crystal structure is complex so it would be helpful to include the crystal size and type of detector (with appropriate citation).

Response: We added the information about crystal size and type of detector in revised the manuscript. The long axis of the crystal was 300 μm and the detector was 'Philatus 3 6M'. (See Materials and Methods)

Comment 2: Then to respect the efforts of other crystallographers, would the authors consider including the following citation given the utility of the program SHELX/C/D/E? (Sheldrick GM, Acta Cryst (2010) D66, 479-485 or Sheldrick GM Acta Crystals (2008) A64, 112-122).

Response: We cited the reference in the revised manuscript, acknowledging the efforts of other crystallographers (See Materials and Methods).

Comment 3: Since the side chain assignment was challenging at 3.2 Angstrom, would the authors state how many Se-Met there were per promoter and how many were identified in the SAD data by ShelxD?

Response: The fragment contained three Met residues in the sequence. However, two Se-Met were identified per protomer in the SAD date by ShelxD. Actually 8 Se sites were identified because the asymmetric unit contained 4 protomers (See Materials and Methods).

Comment 4: Returning the acknowledgement of other crystallographers, the use of Rfree

should perhaps include the Brunger citation: Brunger, Nature(1992) 355:472-475.

Likewise, if ncs restraints were used, as is likely, then citing the appropriate literature would be good: Knowledge-based (Jead, JJ et al. Acta Cryst (2012) D68, 381-390) or Local (Uson, I et al Acta Cryst (1999) D55, 1158-67) should be cited.

Response: We added the references in the manuscript, respecting the efforts of other crystallographers (See Materials and Methods).

Comment 5: The interpretation of the SEC-MALS data in Figure 3 could benefit from some expansion. The C146 construct is clearly in dynamic equilibrium between dimer and tetramer states - evidencing the coiled-coil interaction is fluid. e with reference to Figure 5 the purity of this preparation seems far superior to that used in Figure 5, which then un-necessarily complicates the interpretation in that figure.

The interpretation of the data in Figure 5 could be improved. The gel data in panel A indicate that either proteolytic degradation has occurred or that the cross-linker induces faster migrating species due to intra-chain cross-links. The fact the gel is not properly destained doesn't help. It would help the reader to have these bands in the 35kDa region explained.

Response: We appreciate these helpful comments. In response to the reviewer, we have improved the description of Fig.5. The different conformational states captured after cross-linking are described in the Results part that sulfo-SMPB crosslinker can generate different conformational states by dimerization of the coil 2 fragment at ~30 kDa. It is also possible that because of the length of sulfo-SMBP (1.1 nm), the aliphatic and aromatic linkers not only react with potential lysine and cysteine residues to give an assembly of lamin 300 and coil 2 (R388C), but also induce faster migrating fragments possibly due to the intra-chain crosslinks and non-uniform crosslinking (pages 8-9).

Comment 6: Figure 5B would benefit from more information. The right hand Y-axis is disproportionate to the effect and the interpretation preferred from these data is that K171 is the target Lysine. To me either 180 or 181 are just as good as Lys171. The cross linker has a

1.1nm reach and perhaps the surface geometry of the coiled coils would allow these to be just as involved in this A22 interaction.

Response: We agree with the reviewer's recommendation. We revised the Figure 5B to contain the possible candidates, which bind to R388C of coil2. Fig. 5B was revised to determine the proportion and standard error of the Cys388-bound lysine residues in the lamin 300 fragment by pooled data from the MS/MS analysis of two cross-linked bands, in order to evaluate how significant are the differences between the measurements corresponding to each Lys position, the error bars are given in the graph. We agree with the reviewer's suggestion that positions of Lys180 and Lys181 are as good as the Lys171 target, due to the length of the crosslinker that allows to capture the surface geometric structure of the coiled coils. These changes are included in the Results part (page 9; Fig. 5b-c).

Comment 7: Then the authors might want to embellish the discussion concerning the role of the 42-residue lamin rod feature and how this might impact higher-order assemblies. They might like to consider that the literature contains an example of acytoplasmic IF with just such a feature that appears to incorporate into filaments lacking the rod extension (Gene. 1998 Apr 28;211(1):19-27). It would be helpful to the field to have some speculation here to guide future investigations.

Response: We added the reference to the revised manuscript as suggested by the reviewer (page 12).

Comment 8: An abbreviation list would be helpful - and ensuring that names in full are given at first mention eg Sulfo-SMPB in the text of the manuscript and not hidden in supplementary tables/legends. Please indicate in the Materials and Method whether the His-tag was removed before crystallisation - it has the TEV site after all.

Response: We revised the manuscript as recommended by reviewer (page 8 and materials and methods).

Comment 9: Would the authors like to define the last shell in Table S1 (3.21- ??? A). This is because the rms deviations are very small which could/would have impacted the refinement.

Response: The last shell is 3.21-3.26. We added this information in Supplementary Table 1.

REVIEWERS' COMMENTS:

Reviewer #1 (Remarks to the Author):

. The authors answered to my first questions related to lamin structure and assembly. However, I still do not understand their analysis of the 2 mutants Y45C and L59R (see comment 9). They indeed added the 2 references related to Y45C, but still claim that L59R is associated to muscular dystrophy, without any reference.

. In their Suppl. Figure 6, they show a large panel of mutations at the A11 and A22 interfaces, and they cite: Szeverenyi, I. et al. (2008) and Dittmer & Misteli (2011). In fact, the figure recapitulating the mutations in Dittmer & Misteli already cites Szeverenyi, I. et al. (2008) > this is the only citation to keep. And in this article, the authors give a link to a regularly updated database of lamin mutations: http://www.interfil.org/details.php?id=NM_170707.

. In this database, if you look for Y45C and L59R, you find:

. 1) Y45C: Diseases EDMD2; References Bonne et al, 2000, Scharner et al, 2010

. 2) L59R: Diseases MADA, CMD1A; References: Nguyen et al, 2007, McPherson et al, 2009

. So clearly L59R is not a mutation causing muscular dystrophy.

. And it would be useful to have the list of mutations represented by dots in Suppl. Fig. 6, because now I am not sure that these are really mutations associated to muscular dystrophy, as claimed in the legend.

The fact that L59R is associated to MADA and CMD1A is really interesting to discuss, because it suggests that there might be a relationship between the molecular impact of the mutation (stabilizing vs destabilizing coil1a dimer, disfavoring vs favoring the interaction between coil 1 and coil 2, interface A22) and the disease (muscular dystrophy vs progeria disease for ex). So I don't see why the authors absolutely want to write that L59R is causing a muscular dystrophy.

. Minor point: In the M&M, section - Crystallization and structural determination - « The selenomethionine-labelled lamin protein which cleaved the His-tag », is it correct ?

.

REVIEWERS' COMMENTS:

Reviewer #1 (Remarks to the Author):

Comment 1: The authors answered to my first questions related to lamin structure and assembly. However, I still do not understand their analysis of the 2 mutants Y45C and L59R (see comment 9). They indeed added the 2 references related to Y45C, but still claim that L59R is associated to muscular dystrophy, without any reference.

. In their Suppl. Figure 6 , they show a large panel of mutations at the A11 and A22 interfaces, and they cite: Szeverenyi, I. et al. (2008) and Dittmer & Misteli (2011). In fact, the figure recapitulating the mutations in Dittmer & Misteli already cites Szeverenyi, I. et al. (2008) > this is the only citation to keep. And in this article, the authors give a link to a regularly updated database of lamin mutations: http://www.interfil.org/details.php?id=NM_170707.

. In this database, if you look for Y45C and L59R, you find:

- . 1) Y45C: Diseases EDMD2; References Bonne et al, 2000, Scharner et al, 2010
- . 2) L59R: Diseases MADA, CMD1A; References: Nguyen et al, 2007, McPherson et al, 2009
- . So clearly L59R is not a mutation causing muscular dystrophy.

Response: We were not familiar with the disease names as structural biologist. We thought that CMD stands for cardiac muscular dystrophy. Thank you for providing us the opportunity to correct it. We revised the manuscript accordingly, and added the references to the revised manuscript.

“To gain insights into the laminopathies at the molecular level, we selected two mutations Y45C and L59R, which are related to muscular dystrophy^{37, 38} and dilated cardiomyopathy^{13, 39, 40} respectively.”

Comment 2: And it would be useful to have the list of mutations represented by dots in Suppl. Fig. 6, because now I am not sure that these are really mutations associated to muscular dystrophy, as claimed in the legend.

Response: The mapping mutations in Supplementary Fig. 6a. are associated to laminopathies

with preferential involvement of skeletal and cardiac muscles. The detailed information was described in the reference (Figure 6, red; Dittmer et al, 2011). We added the mutation residues in Supplementary Fig. 6a to address the reviewer's comment.

Comment 3: The fact that L59R is associated to MADA and CMD1A is really interesting to discuss, because it suggests that there might be a relationship between the molecular impact of the mutation (stabilizing vs destabilizing coil1a dimer, disfavoring vs favoring the interaction between coil 1 and coil 2, interface A22) and the disease (muscular dystrophy vs progeria disease for ex). So I don't see why the authors absolutely want to write that L59R is causing a muscular dystrophy.

Response: We don't have any intention on L59R mutation. We were not familiar with the disease names as structural biologist. We thought that CMD stands for cardiac muscular dystrophy. Thank you for providing us the opportunity to correct it. We revised the manuscript accordingly. We think that this correction would increase importance of our study.

Comment 4. Minor point: In the M&M, section - Crystallization and structural determination - « The selenomethionine-labelled lamin protein which cleaved the His-tag » , is it correct ?

Response: We corrected the sentence as following:

“The selenomethionine-labelled lamin protein (7 mg/mL) whose cleaved the His-tag and it was crystallized in a precipitation solution containing 0.1 M Tris-HCl (pH 8.5), 0.9 M lithium chloride, and 7% (w/v) PEG 8000 using the hanging-drop vapour diffusion method at 14°C.”